



# Does Belgian Holocene speleothem records solar forcing and cold events?

Mohammed Allan[1], Adrien Deliège[2], Sophie Verheyden[3], Samuel Nicolay2, Yves Quinif[4],

Nathalie Fagel[1]

[1]AGEs, Département de Géologie, Université de Liège, Allée du 6 Août, B18 B-4000, Liège, Belgium

[2] Institut de Mathématique, Université de Liège, Allée de la Découverte 12, B37 B-4000 Liège, Belgium

[3]Politique scientifique fédérale, 231 Avenue Louise, Bruxelles, 1000- Brussels, Belgium,

[4]Université de Mons, Rue de Houdain9, B-7000 Mons, Belgium

Correspondence to: M. Allan (mohammed.allan@ ulg.ac.be)

**Abstract.** We present a decadal-centennial scale Holocene climate record based on trace elements contents from a 65 cm stalagmite ("Père Noël") from Belgian Père Noël cave. Père Noël (PN) stalagmite covers the last 12.7 ka according to U/Th dating. High spatial resolution measurements of trace elements (Sr, Ba, Mg and Al) were done by Laser-Ablation Inductively Coupled Plasma Mass Spectrometry (LA-ICP-MS). Trace elements profiles were interpreted as environmental and climate changes in the Han-sur-Lesse region. Power spectrum estimators and continuous wavelet transform were applied on trace elements time series to detect any statistically significant periodicities in the PN stalagmite. Spectral analyses reveal decadal to millennial periodicities (i.e., 68-75, 133-136, 198-209, 291-358, 404-602,912-1029 and 2365-2670 yr) in the speleothem record. Results were compared to reconstructed sunspot number data to determine whether solar signal is presents in PN speleothem. The occurrence of significant solar periodicities (i.e., cycles of Gleissberg, de Vries, unnamed 500 years, Eddy and Hallstat) supports for an impact of solar forcing on PN speleothem trace elements contents. Moreover, several intervals of significant rapid climate change were detected during the Holocene at 10.3, 9.3-9.5, around 8.2, 6.4-6.2, 4.7-4.5, and around 2.7 ka BP. Those intervals are similar to the cold events evidenced in different natural paleoclimate archivers, suggesting common climate forcing mechanisms related to changes in solar irradiance.

**Keywords:** Holocene, climate change, solar forcing, cold events, speleothems, wavelet analysis, trace elements, paleoclimate time-series.

## 1. Introduction

The Holocene period appears as a relatively stable climatic period compared to Quaternary glacial/interglacial variations. However several high-resolution studies have emphasized many climatic oscillations over the last 11500 yr (e.g. Mayewski et al., 2004; Wanner et al., 2008, 2011). The Holocene climate variability was tentatively attributed to orbital forcing, volcanic, and/or solar activity (Wanner et al., 2008). Many climatic proxies (e.g., $\delta^{18}O$, $\delta^{13}C$, $\Delta^{14}C$, $^{10}Be$, REE content, pollen, dust, humification, element traces) have been measured in different geological





archives: in peat (e.g., Allan et al., 2013; Marx et al., 2009, 2011; Shotyk et al., 2001, 2002), speleothems (e.g.,

Fairchild and Baker, 2012; McDermott et al., 2011; Genty et al., 2003; Verheyden et al., 2008), ice cores (e.g., Delmonte et al., 2002; Gabrielli et al., 2010; Thompson et al., 2000), and in lake sediments (e.g., Magny M., 2004), to reconstruct the past climatic changes over the Holocene. Based on statistical analysis of proxy time series, Wanner el al., (2011) showed six specific cold events relapses in Northern Hemisphere during the Holocene (8.2 ka, 6.3 ka, 4.7 ka, 2.7 ka, 1.5 and 0.5 ka BP). These episodes have alternately been attributed to freshwater forcing, volcanic activity,

and/or changes in Atlantic Meridional Overturning Circulation, and low solar variability (Wanner et al., 2011). Many studies have reported evidence for solar-forcing on Holocene climate change across a range of continental and marine archives (e.g., Bond et al., 2001; Soon et al., 2014). Holocene solar activity variation showed several solar cycles on different temporal scales, Schwabe cycle (11 yr), the Gleissberg cycle (70-100 yr), Hale cycle (130 yr), de Vries cycle (200-210 yr), Eddy cycle (1000 yr), Bray or Halstatt cycle (2200-2400 yr). A number of climate proxies have been

used to investigate solar forcing including geochemical and biological records from tree rings (Muraki et al., 2011), lake sediments (Kern et al., 2012), peatland (Turner et al., 2016) and speleothems (Niggemann et al., 2003).

During the last decades, speleothem studies have enhanced our knowledge of Holocene continental climate, increasing the chronological resolution and extending the period covered (Drysdale et al., 2009; Fleitmann et al., 2009; Genty et al., 2003; Wang et al. 2008). Their ability to be U-series dated with a precision of 0.5% (Edwards et al., 1987)

combined with the possibility of layer counting when seasonal layers are present (Verheyden et al., 2006; Genty and Quinif, 1996), and even more if luminescent layering (Shopov et al., 1994) is present, makes speleothems a powerful tool to refine the chronology and increase the resolution of climate reconstructions. Speleothems contain several already well-studied climate proxies such as $\delta^{18}O$, $\delta^{13}C$, strontium isotopes and compositions (Fairchild and Baker, 2012; McDermott et al., 2011; Mangini et al., 2005; Mattey et al., 2008; Zhou et al., 2009; Goede et al., 1998). In

addition, speleothem trace elements compositions represent a large proportion of speleothem proxies, used to reconstruct paleoenvironmental conditions. Variations in solar activity have been well documented in Holocene speleothems archives, by identifying changes in color or isotopic composition (e.g., Niggemann *et al.*, 2003; Cosford et al., 2009; Martín-Chivelet et al., 2011) or variations in the thickness of speleothem layers (e.g., Genty and Quinif, 1996; Frisia et al., 2003; Munoz et al., 2009). As previously investigated in the Han-sur-Lesse cave from the

same karstic system as Père Noël cave, and presented in Genty and Quinif (1996), the Han-stm-1 stalagmite has shown the presence of successive laminae forming very regular cycles of 11 years separated by a thick dark compact lamina, which is similar to the sunspot solar cycle. Until now, the potential of trace elements content in speleothems for recording solar activity variations is largely unexplored.

In previous low resolution studies, Verheyden et al. (2008, 2014) already observed that the PN stalagmite displays

important variations in growth rate, geochemical (Mg/Ca and Sr/Ca) and isotopic (O, C) content along its longitudinal axis between 12.7 and 1.8 ka. Changes in isotopic equilibrium conditions were interpreted as due to changes in cave humidity and linked to changes in recharge, ie. precipitation minus evapo-transpiration. Changes in trace element content were interpreted as due to changes in water residence time linked to changes in water availability (Verheyden et al., 2008). In this study, by using high-resolution trace elements content (Ba, Sr, Mg and Al), we reconstruct climate

changes that occurred in Northwestern Europe during the Holocene. We apply spectral and wavelet analysis of the



trace elements time-series of the PN stalagmite to identify solar-type periodicities in the stalagmite record. Moreover, the sunspot reconstruction of Solanki et al., (2004) was compared with the PN times series to identify eventual solar cycles in the PN proxies and as such determine the possibilities for the study of solar activity changes over the Holocene.


### 2. The state of the art of trace elements in speleothems

To understand the significance of the elemental variations in speleothems (i.e., Ba, Sr, Al and Mg), it is necessary to investigate the potential sources of the chemical elements and their transfer through the karst system. Then, the relations between stable isotopes and trace elements are characterized and may be related to paleoclimate changes.

Elemental trace contents may be derived from different sources such as natural and/or anthropogenic dust (local or distal) supplies (Allan et al., 2015; Dredge et al., 2013; Fairchild et al., 2000; Frisia et al., 2005; Wynn et al., 2008). They may also derive from the host rocks and adjacent soil cover (Tooth and Fairchild, 2003; Hartland et al., 2012; Treble et al., 2016). The trace elements could be transported along the soil profile by colloids or in dissolved forms depending on the element. The trace elements might show different forms of mobility (in dissolved and particulate

form) and speed of migration in soils, depending on soil characteristics and processes (e.g., pH, organic acid content, redox conditions, leaching, ion exchange, temperature). Finally, the karst hydrology may also play a role in the speleothem trace element content (Fairchild et al., 2000; Tooth and Fairchild, 2003). Karstic processes include organic matter decomposition, dissolution, desorption and adsorption of colloids, Prior Calcite Precipitation (PCP), incorporation into carbonate minerals (e.g., Fairchild and treble, 2009).

Previous studies have shown that the variations of trace element contents in a speleothem may provide information about changes in soil condition, water temperature, paleorainfall and hydrological conditions (e.g., Roberts et al., 1999, Fairchild et al., 2000; Fairchild and Treble, 2009). In temperate regions, trace element such as Ba, Sr and Mg tend to increase during drier periods when residence times are longer (Huang et al., 2001; Verheyden et al., 2000). Many studies performed on speleothems from different environmental settings reveal that both hydrological (e.g.,

amount of precipitation, groundwater residence time, rock-water interaction) and/or growth-related processes can affect trace element concentrations (e.g., Verheyden et al., 2000; Treble et al., 2003; Tremaine and Froelich, 2013). However, the link between trace element content in speleothems and climatic conditions can vary between caves settings due to differences in chemical composition of bedrock, groundwater movement, soil thickness and/or climatic conditions. Consequently, a thorough comparison between speleothem trace element content and another proxy (e.g.,

combining stable isotopes with trace elements and/or petrographic information and/or growth-rate) is important in order to minimize the uncertainties associated with stable isotope data interpretation (Fairchild et al., 2006; McDermott, 2004; and Verheyden et al., 2001).

### 3. Material and methods

PN stalagmite, 65 cm long, was taken in 2000 from the Belgian Père Noël cave (50.0°N, 5.2°E, 230 masl) in the Ardenne massif (Fig.1). The cave located about 200 km inland in a temperate maritime climate. The cave opened in the Devonian (Givetian) Fromelenne Formation with subvertical limestone beds (Delvaux de Fenffe, 1985) that reach a thickness of 70 m above the cave (Deflandre, 1986). The water entering the cave consists of local rain only, seeping



directly through the overlying limestone. An approximately 40-cm-thick woodland soil covers the host rock limestone.

The mean annual precipitation of the nearby meteorological station (Lessive, 3 km NE of the cave), measured over the period 1980-1998, is 826 mm. The mean temperature inside the cave is 9 °C (varies between 8.5° and 9.2 °C) wichcorresponds to the mean annual temperature of the air (9.2°C) at the nearby Lessive. A detailed description of climate, geology, and hydrological conditions were published in Verheyden et al. (2000, 2008, 2012).

The PN stalagmite was dated by TIMS U-series dating (10 dates) at the Department of Earth Sciences at Open

University, UK. Other dates (6 dates) were done by NEPTUNE Multi-Collector Inductively Coupled Plasma Mass Spectrometry (MC-ICP-MS) at the Laboratoire Géosciences Environnement Toulouse (GET) in France and one date was obtained in the Earth Science Department of the University of Minnesota on a Thermo-Scientific Neptune-Plus multi-collector inductively coupled plasma mass spectrometer (MC-ICP-MS). The procedures that were followed for uranium and thorium chemical separation and purification described in Edwards et al. (1987) and Cheng et al. (2009a,

2009b). Dating results are summarized in Table 1, with ages given in years before 1950. Errors given in the table ($2\sigma$) depend on the U and Th content and age of the sample. The elementary geochemical composition (Ba, Sr, Mg and Al) of PN stalagmite was determined by Laser Ablation Inductively Coupled Plasma-Mass Spectrometer (LA-ICP-MS) mounted with an ESI New Wave UP-193FX Fast Excimer ArF laser of 193 nm at the Royal Museum for Central Africa (Tervuren, Belgium). Spots were made of 50 μm in diameter and spaced at 300-1000 μm intervals. The age

model reveals a growth rate varied between 0.02 and 0.65 mm.yr$^{-1}$. I calculted consequently the analysis of 50 micrometers each 300-1000 micrometers interval corresponds to a sample of 1 to 3 years every 0.5 to 50 years. Detection limits are calculated from the intensity and standard deviation measurements of the blank. The limits of quantification range between 0.1μg g$^{-1}$ for Sr, Al and Ba, and 2.5μg g$^{-1}$ for Mg. Certified reference materials (NIST 610, NIST 612, NIST 614, MACS-1, and MACS-3) were analyzed with each series of samples, in order to determine

the precision and accuracy of analytical procedures. Comparison between reference values and measured values were satisfactory (Table 2) within 65-98 %. For Ba, Sr, and Mg, the reproducibility was higher than 76%. The lowest value was observed for Al (65% for MACS-1 standard). To investigate the solar forcing controls on trace elements content in the PN stalagmite, we compared the concentrations of Ba, Sr, Mg with sunspot number (solar activity-Solanki et al., 2004) and the temperature recorded from Mekelermeer core in Netherlands (Bohncke P., 1991). Continuous wavelet

transform was applied on the trace element time-series data to detect any significant periodicities. These were obtained as the local maxima of the wavelet spectrum (see supplementary data).

## 4. Results

The age-depth model (Fig. 2) indicates that the PN stalagmite was deposited between 12.7 and 1.8 ka BP. The

internal longitudinal section of the stalagmite presents a succession of brown parts and milky white parts along its longitudinal axis (Fig. 2). No clear hiatus is observed in the stalagmite. The longitudinal section also reveals variations in the stalagmite diameter along its longitudinal axis. The growth rate varied between 0.02 and 0.65 mm.yr$^{-1}$ with the highest growth rates (0.09 to 0.65 mm.yr$^{-1}$) observed between 7.8 and 6.3 ka BP (i.e., between 37 and 12 cm). The lowest growth rates evidenced between 6.3 and 1.8 BP ka.





The Ba, Sr, Mg, and Al records in PN are composed of nearly six-hundred independent points. Profiles of Mg, Sr, Ba and Al in PN stalagmite are very similar (Fig. 3). The values (in µg g⁻¹) range from 5 to 65 for Ba, 20 to 160 for Sr, 1000 to 17600 for Mg and from 0.2 to 35 for Al. High trace elements contents are observed at different depths:  from 540 to 455 mm (10.7-9.2 ka BP), 390 -370 mm (8.2-7.9 ka BP), between 300 and 140 mm (7.2-6.2 ka BP), and from 25 mm (3 ka BP) to the top of PN stalagmite at 2.02 ka BP (Fig.3, 4). In the low trace element content zones, the Al

content was below the limit of detection (< 0.1 µg g⁻¹). Trace element results are closely correlated (r = 0.48-0.77, Fig. 3), implying that similar processes influence their concentration changes, over all timescales. Strontium concentration averages approximately 2-4 times the Ba concentration, throughout the record. Trace element data was resampled at constant time interval to perform spectral analyses (more details in supplementary data). Significant periods of 71, 136, 198, 291, 497-602, 1029, 1365, 1706, 1861 and 2315 are observed in Ba time-series. Strontium also shows

significant periodicities at 75, 133, 198, 281, 404-634, 960, 1280, 1365, 1855, and around 2630 yr. In addition, Mg data exhibit peaks around 68, 133, 209, 325, 500, 602-706, 967, 1533 and around 2365 yr. All observed periodicities display some similarities with solar variability, presented by sunspot number, which shows significant cycles at 68, 87, 136, 225, 352, 525, 975, 1462, 2275 yr (fig.5) .

Stable isotopes data (δ¹⁸O) from PN stalagmite reported in Verheyden et al. (2008). The δ¹⁸O composition of

PN varies between −4.3 and −6.4‰ (VPDB) with a mean of -5.4‰ (Fig. 4). It displays a long term increasing trend from lower values (-6‰) between 12.7 and 10.7 ka BP to -5.3 ‰ between 10.7 and 7.5 ka BP and to higher values (>-5.0‰) between 6.3 and 1.8 ka BP. Lower values are observed at 10.7, between 9.5 and 9, between 8.3 and 6.2, at 5.5 ka BP and between 4.3 and 3.4 ka BP with a general increase until the end of speleothem deposition. The δ¹⁸O isotopic composition of the calcite in the Père Noël cave is largely controlled by water availability (drip rate) in the cave.

Changes in isotopic equilibrium conditions are driven by the changes in cave humidity and linked to changes in precipitation and evapo-transpiration (Verheyden et al., 2008).

## 5.  Discussion

### 5.1. Processes controlling trace element contents

In previous studies, Verheyden et al., (2000, 2008, 2012) interpreted changes in the different PN cave parameters and speleothem proxy-data in terms of changes in environment and/or climate. Higher Mg/Ca and Sr/Ca ratios were previously observed during the summer season, characterized by longer water residence times ( Verheyden et al., 2008) and associated to lower water availability. Since water residence time is related with water availability, trace elements in PN speleothem probably register changes in effective precipitation, i.e., precipitation minus evapotranspiration. In

this study, statistically significant positive correlations ($r_{(Sr, Ba)}$=0.77, $r_{(Mg, Sr)}$=0.72, $r_{(Mg, Ba)}$=0.48) are found between Sr, Ba and Mg over the Holocene period suggesting either a common or strongly related controlling process. Aluminum can be used to determine the variations in the detrital (non-carbonate) content in speleothems; these particles may be transported during periods of intense precipitation, which results in high drip rates (White, 1997; Schimpf et al., 2011). In PN stalagmite, Al concentration was higher than the detection limit and its maximum values coincide with maxima

in the other investigated trace elements investigated (Fig.4). The positive correlations of Al with Mg, Ba and Sr (r = 0.45) suggest that Al content is controlled by the same process than the other trace elements. The δ¹⁸O profile has a



positive correlation with Sr, Ba, and Mg (r= 0.45-0.70), suggesting a common control. Changes in element traces content in PN stalagmite are interpreted as due to changes in water residence time linked to changes in water availability. This explains the covariation over much of the Holocene period of the geochemical parameters in the PN

speleothem.

### 5.2. Spectral and wavelet analyses

First spectral analysis demonstrates that trace element time-series contain significant periodicities on decadal to millennial times scales (Fig.5). The most frequently occurring periodicities were those at 68-75, 133-136, 198-209, 291-358, 404-602,912-1029 and 2365-2670 yr (Fig.5). The temporal stability of those periodicities is confirmed by

wavelet results (Fig. 6a, b, c).

Second trace element PN time-series are compared with available temperature record from Mekelermeer core in the Netherlands (Bohncke J., 1991) in order to evidence a regional temperature influence on the PN record. Both records reveal significant correlations.

Third we test PN (and temperature) time-series for solar forcing by applying wavelet approach. Solar forcing was

represented by the number of sunspots (Solanki et al., 2004). Wavelet analyses of Ba, Sr and Mg and temperature time-series with reconstructed sunspot number time series reveal significant common periodicities within (63–80, 133-140, 198–220, 514-561, 912-1029 and 2245-2670 yr-Fig.6, supplementary data). Those periodicities coincide with known solar cycles (e.g., Steinhilber et al., 2012; McCracken et al., 2013) and are in agreement with decadal-centennial scale variability in Holocene climatic records from widely dispersed geographic regions. For example, the

63–80 yr interval is similar to the Gleissberg cycle (70-100 yr) whereas the 133-140 yr frequency interval corresponds to the Hale cycle (130 yr). The 198–220 yr cycle of PN record is close to the de Vries (200-210 yr) solar cycle (Steinhilber et al., 2012). De Varies's cycle was already observed in numerous palaeoclimate records all over the world (Wanger et al., 2001; Duan et al., 2014; Czymzik et al., 2016). There are also strong variability around 281-325 yr and 535 yr (514-561) periods that correspond to solar variability that reported as "350 and 500 unnamed cycles"

(Steinhilber et al.,, 2012). These cycles were also detected in numerous other palaeoclimatic records worldwide (e.g., Soon et al., 2014; Chapman and Shackleton, 2000). The 960-1029 yr cycle of the speleothem could match the Eddy cycle (1000 yr) as recognized in both ice cores (Stuiver et al., 1995) and marine sediments (e.g., Chapman and Shackleton, 2000). The trace elements PN records show small peaks between 1280 and 1533 yr (Fig. 5). This periodicity is identical to the Bond cycle (1470 ± 500 yrs) detected from ice rafted debris in North Atlantic sediment

cores (Bond et al., 2001). Finally the 2245-2670 yr cycle, could be related to the Bray or Halstatt cycle (2200-2400 yr) that is recognized in other palaeoclimatic records (e.g., kern et al., 2012). The significant common periodicities between trace elements PN records and Solar records suggest that solar variability influenced PN trace elements content on decadal to millennial time scales. Since the changes of trace elements in the PN speleothem were formerly demonstrated as due to changes in recharge, i.e. effective precipitation, the study suggests that variations in solar

activity may be a significant forcing on either the precipitation or on the soil activity  or vegetational activity intensity. This does not necessarily mean that solar forcing was the main source of all Holocene climate variability since different dynamical processes, such as explosive volcanic eruptions, fluctuations of the ocean thermohaline circulation or internal feedbacks, might also have played an important role (Wanner et al., 2008).



### 5.3. Relation between trace element records and known Holocene climate events

The PN records indicate that the maxima of trace elements concentrations coincide with the cold events (Fig. 4) defined in Wanner et al. (2011), which partly coincide with the events of Bond et al. (2001). The higher positive correlations between trace elements concentrations (r=0.5-0.9) observed during cold periods which also suggest common control factors. In marine sediments Bond et al., (1997, 2001) revealed 1470 yrs± 500 Holocene cycles. Nine Bond cycles were detected over the Holocene and peaked around 0.4, 1.4, 2.8, 4.3, 5.9, 8.1, 9.4, 10.3 and 11.1 ka BP

(Bond et al., 1997, 2001; Wanner and Bütikofer, 2008b). These events, attributed to lower solar activity, most probably triggered the slowdown of the thermohaline ocean circulation over the North Atlantic and European regions (Renssen et al., 2007). Figure 7 shows a comparison of the trace elements concentrations in the PN stalagmite and the frequency of warm/cold and wet/dry events in the Northern Hemisphere (Wanner et al., 2014). In the PN stalagmite, we identify several periods of dry climate, these are centered at 10.5, 9.3, 8.2, 6.3, 5.4, 4.6 and around 2.7 ka BP, and

their duration ranges from 100 to 400 yr (Fig. 7). These intervals characterized by higher trace elements contents in PN record were interpreted as dryer periods.

### 5.3.1. Younger Dryas and Early Holocene

The PN stalagmite starts its growth on the underlying limestone block at 12.7 ka BP, indicating favorable climatic and environmental conditions for stalagmite growth. The oldest part of the PN stalagmite, from 12.7 to about 12.1 ka

BP, is characterized by the lowest contents of trace elements and low $\delta^{18}$O, suggesting wet conditions (Fig.4). The larger diameter of the stalagmite compared to its mean diameter suggests that there is enough water to flow down on the flank of the stalagmite and to precipitate calcite. The inception of speleothem growth is in agreement with the beginning of the cold Younger Dryas (YD) event (12.9-11.7 ka BP). All proxies measured in the PN stalagmite suggest that a humid period occurred between 12.7 and 12.1 ka BP. This is in agreement with the warm climate during

the early YD (12.9–12.15 ka BP) found in lake sediments from Northern Spain, Norway and Western Germany (Baldini et al., 2015; Bakke et al., 2009; Brauer et al., 2008). The high-resolution records from those three lakes suggested that warming climate associated with the early YD may be related to local climate change (Lane et al., 2013; Baldini et al., 2015). Our high-resolution trace elements from PN stalagmite consolidate the concept that the early YD (12.9–12.15 ka BP) was warm in SW and NW Europe. The period, from 12.1 to about 11.7 ka BP, is characterized by

a gradual increase of trace element concentrations, suggesting a climate change from wet to dry conditions (Fig.4, 7). The $\delta^{18}$O values increase from -6.3 to -5.5 ‰ suggests relatively dry conditions in agreement with trace element concentrations (Fig.4). European archives (lake and speleothem records) indicate that a cooler and/or drier climate characterized the YD termination (12.1-11.7 ka BP) (e.g., Genty et al., 2006; Von Grafenstein et al., 1999). This rapid climate shift attests for the major influence of the North Atlantic Ocean circulation on the YD/Holocene climate

transition (Pearce et al., 2013; Baldini et al., 2015).

During the Early Holocene, i.e. before 7 ka BP, the amplitude of the frequency of warm/cold or dry/ wet events was from the rapidly melting ice sheets of the Northern Hemisphere (Carlson et al., 2014). The Holocene started about 11.7 ka BP with a transition from a cool YD to a wet and/or warm climate state. In the PN stalagmite, low trace element contents suggest that wet conditions persisted at the beginning of the Holocene (Fig.4). The $\delta^{18}$O values decrease from

-5.5 to -6 ‰ suggests relatively wet conditions in agreement with trace elements concentrations. PN speleothem



suggests a wet Early Holocene with dryer conditions from 10.7 to 10.3, at 10.0, at 9.7, at 9.2 and from 8.5 to 8.2 ka BP (Verheyden et al., 2014). Trace element contents in the PN stalagmite display significant variability during the Early Holocene, with three maxima observed around 10.5, from 9.5 to 9.2 and around 8.2 ka BP (Fig. 4, 7). The positive correlations (r=0.47-0.98) were observed between trace elements concentrations during those periods which confirms

common control factors. The covariance between geochemical proxies support, as explained in Verheyden et al. (2008), their interpretation in terms of dry-wet changes with higher trace elements and higher $\delta^{18}O$ values linked to drier conditions. The drying is in agreement with a decrease in speleothem growth rate and diameter. Moreover, a denser calcite is deposited at 10.4 ka, between 9.5 and 9.2 and between 8.4 and 8.2 ka BP (Fig.4). The highest concentrations of trace elements in PN correspond to low sunspot numbers and to cold periods (Fig. 4), as defined in

Bond et al., (2001). During the three intervals (i.e., around 10.5, between 9.5 and 9.2 and between 8.4 and 8.2 ka BP), the wavelet spectrum of trace elements shows that the solar imprint is well present in the PN stalagmite records during the Early Holocene (Fig.6). Those three intervals may correspond to Bond cycles that caused by the lower solar activity additionally triggered slowdown of the thermohaline ocean circulation in the North Atlantic and European regions (Renssen et al., 2007). Continuous cyclic periods of around 500 and 1000 yrs are presented during the interval

10.7-10.3 ka BP. For the intervals from 9.5 to 9.2 and between 8.4 and 8.2 ka BP, continuous cyclic periods of around 80, 130, 500, 1000, 1300-1500 yr are present. Ice, speleothems and sediment records from the North Atlantic region demonstrate that the Early Holocene period was relatively stable and interrupted by periods of short-lived cooling such as: 10.3 ka (Bjorck et al., 2001), 9.3 ka (e.g., Boch et al., 2009; Rasmussen et al., 2007; von Grafenstein et al., 1999) and 8.2 ka event (e.g., Alley et al., 1997; Ellison et al., 2006). Solar variability and/or changes in Atlantic Meridional

Overturning Circulation have been proposed as causes of climate changes in the Early Holocene (Björck et al., 2001; Marshall et al., 2007; Plunkett and Swindles, 2008). A cold event around 8.2 ka BP has been recorded in the North Atlantic and Northern Europe (Alley et al., 1997; Barber et al., 1999; Clark et al., 2001). This event has been well recorded in PN stalagmite and discussed in detail in Verheyden et al., (2014). The correlation between trace elements and sunspot number suggests that periods of lower intensity of solar irradiation were probably accompanied by drier

climate in NW Europe. However, solar forcing cannot alone explain the variability of the Early Holocene signal. Previous studies in ice cores (e.g. Alley et al., 1997) or lake sediments (e.g., Prasad et al., 2006), demonstrated that the climate changes in Western Europe and Greenland are similar during the Early Holocene and may be caused by changes in solar activity, and changes in North Atlantic thermohaline circulation.

### 5.3.2.    Mid and Late Holocene

The interval from 8.1 to about 7 ka BP is characterized by low contents of trace elements and low $\delta^{18}O$ values suggesting relatively wet conditions. This phase coincides with the interval of maximum frequency period of warm and wet events detected in the Northern Hemisphere (Wanner et al., 2014) (Fig.7). The interval between about 7 and 4.2 ka BP "called Holocene Climate Optimum" is characterized by higher summer temperatures in the Northern Hemisphere (e.g., Alverson et al., 2003). High trace elements values were detected in PN stalagmite during three

intervals, from 6.4 to 6.2, 5.55 to 5.4 and from 4.7 to 4.5 ka BP (Fig.4). The period between 7.3 and 6.0 ka BP corresponds to the warmest period of the Holocene in Greenland (Jonhsen et al., 2001). From 6.4 to 6.2 ka BP a high positive correlation (r=0.45-0.95) was observed between element trace concentrations confirming dryer conditions. The drying is in agreement with a decrease in the speleothem growth rate and diameter. The high concentrations of



trace elements in PN correspond to low temperature and sunspot number (Fig. 4). This phase coincides with the period

of maximum frequency of dry events detected in the Northern Hemisphere (Wanner et al., 2014) and with a cold event as defined in Wanner et al., (2011).  Between 5.5 and 5.4 ka BP, the trace elements (high values) display an anticorrelation with temperature and sunspot number (low values) suggesting relatively dry conditions (Fig.4). During this period, trace elements present frequencies corresponding to the Gleissberg cycle (70-100 yr), Hale cycle (130 yr), de Vries cycle (200-210 yr), 500 unnamed cycle and Eddy cycles (1000 yr) (Fig.6).

High trace elements and $\delta^{18}O$ values were detected in the PN stalagmite from 4.7 to 4.5 ka BP corresponding to a dry period. The highest positive correlation (r=0.78-0.97) observed between trace element concentrations during this period confirms dry conditions. The period between 4.7 and 4.5 ka BP corresponds to the cold event (4.6-4.8) defined by Wanner et al., (2011) and coincides roughly with Bond events. This phase coincides with the period of decrease in frequency of wet events and increased of dry events detected in the Northern Hemisphere (Fig.7). In addition, our

spectral analysis shows the prominence of 63–80, 133-140, 198–220, 514-561, and 912-1029 year periodicities in the PN stalagmite (Fig.6).The periodicities found here suggests evidence for solar-forced climate change because they match the ranges of cycles in solar reconstructions. From 4.5 to 3.1 ka BP, general climatic conditions favor the speleothem deposition. Low values for trace elements and for $\delta^{18}O$ indicate relatively wet and temperate/warm conditions.

General dry conditions as suggested by PN proxies were observed from 3 ka BP to the end of the speleothem deposition at 1.8 ka. The final dry phase may be related to the cool event showed by Wanner et al. (2011) that occurred between 3.3 and 2.5 ka BP and corresponds with Bond event (Fig. 4).This dry period also coincides with low temperature and solar activity (Fig. 4). This period was caused by an abrupt decrease of solar activity and has been reported from different proxies in Greenland and Europe (e.g., van Geel et al., 1996; Blaauw et al., 2004; Plunkett and

Swindles, 2008; O'Brien et al, 1995).  The overall good agreement of the timing of the dry/wet or cold phases recorded in the PN stalagmite with the timing of the Wanner and Bond events (Wanner et al., 2001, 2014; Bond et al., 1997, 2001) confirms the potential of the PN speleothem to reconstruct the Holocene paleoclimate.

**6.  Conclusion**


      We have shown that the high-resolution trace element records obtained by LA-ICP-MS from the PN stalagmite provide a detailed paleoclimate and/or paleoenvironment record of Northwestern Europe through the Holocene. The strong covariation of trace elements (Ba, Sr, Mg and Al), and with $\delta^{18}O$, confirms a common or strongly related controlling process. Based on trace element time-series we demonstrate that several events at 10.3, 9.3-9.5, around 8.2,

6.4-6.2, 4.7-4.5, and around 2.7 ka BP alternate with periods of relatively stable and wet/warmer climate. These intervals coincide with the cold events defined in marine and continental archives. The trace element time-series of the PN speleothem reveals a significant correlation with sunspot number records, suggesting some solar forcing in the PN trace elemental records. This observation is confirmed by wavelet analyses that reveal common solar periodicities (Gleissberg cycle, de Vries cycle, unnamed 500 year, Eddy cycles, and Hallstatt cycle) in agreement with those

recognized in the North Atlantic marine cores and the Greenland ice cores, as well as some other terrestrial Holocene records, suggesting common forcing mechanisms. Our study, based on high-resolution LA-ICP-MS analyses,





emphasizes that speleothem trace element profiles may be considered as a new solar activity proxy on decadal-centennial timescales over the Holocene.

**Author contribution:** M. Allan, S. Verheyden and N. Fagel designed the experiments and A. Deliège and S. Nicolay carried out Statistical data analysis, Y. Quinif was a part of the speleothem coring team. M. Allan prepared the manuscript with contributions from all co-authors.

**Competing interests:** The authors declare that they have no competing financial interests.

**Acknowledgement:** We thank all the persons who contribute to this work and especially Laurence Monine from
the *Royal Museum* for *Central* Africa (LA-ICP-MS analyses), prof. Denis Scholz from Johannes Gutenberg University Mainz (Age model). This study was funded by the FNRS (HOPES project, N° 23612574).

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





Table 1: Mass-spectrometric U/Th age data for the PN speleothem.

| Distance from top of stalagmite (mm) | U(ppm) | $^{234}U/^{238}U$ | $^{230}Th/^{232}Th$ | $^{230}Th/^{234}U$ | Age in years BP 1950 | error +/-2σ | Laboratory |
|---|---|---|---|---|---|---|---|
| 5 | 0,234 | 2,366 | 30 | 0,0185 | 2024 | 11 | Open Univ-UK |
| 45 | 0,297 | 2,391 | 59 | 0,0354 | 3907 | 16 | Open Univ-UK |
| 85 | 0,223 | 2,438 | 58 | 0,0518 | 5585 | 39 | GET-FR |
| 100 | 0,214 | 2,434 | 207 | 0,0501 | 5911 | 47 | GET-FR |
| 115 | 0,199 | 2,466 | 980 | 0,0585 | 6515 | 37 | Open Univ-UK |
| 170 | 0,181 | 2,462 | 335 | 0,0723 | 6461 | 48 | GET-FR |
| 190 | 0,168 | 2,497 | 1216 | 0,0613 | 6843 | 37 | Open Univ-UK |
| 250 | 2,781 | 2,496 | 2345 | 0,0641 | 7164 | 22 | Open Univ-UK |
| 307 | 0,199 | 2,492 | 1058 | 0,1066 | 7151 | 51 | GET-FR |
| 365 | 0,228 | 2,506 | 1902 | 0,0701 | 7853 | 45 | Open Univ-UK |
| 395 | 0,242 | 2,514 | 1401 | NA | 8232 | 39 | Open Univ-UK |
| 425 | 0,238 | 2,552 | 1042 | NA | 9051 | 27 | Minnesota Univ-USA |
| 435 | 0,304 | 2,578 | 724 | 0,0816 | 9185 | 21 | Open Univ-UK |
| 490 | 0,269 | 2,595 | 1241 | 0,0888 | 9757 | 63 | GET-FR |
| 520 | 0,253 | 2,654 | 299 | 0,0912 | 10303 | 55 | Open Univ-UK |
| 635 | 0,179 | 2,729 | 161 | 0,1112 | 12671 | 71 | Open Univ-UK |
| 635 | 0,203 | 2,724 | 344 | NA | 12648 | 66 | GET-FR |

Decay constants used to calculate activity ratios from measured ratios are defined as follows: $\lambda^{238}U=1.551*10^{-10}$, $\lambda^{234}U=2.835*10^{-6}$, $\lambda^{230}Th=9.915*10^{-6}$, $\lambda^{232}Th=4.948*10^{-11}$. Age uncertainties are reported at the 2σ level.



Table 2 : Comparison between reference and measured values for five certified reference materials.

| Standard | Element (µg.g⁻¹) | Ba | Sr | Mg | Al |
|---|---|---|---|---|---|
| NIST 610 | Reference values | 473 ±4 | 497,4 ±3,3 | 465±12,4 | 10800 |
| | Mean measured values (n=7) | 565 ±133 | 658 ±139 | 625±65 | 11458±215 |
| | Reproducibility | 76 | 79 | 90 | 98 |
| NIST 612 | reference values | 37,74 ±1 | 76,15 ±0,5 | 77±4 | 11170 |
| | Mean measured values (n=24) | 45,1 ±9,7 | 82,6 ±6,6 | 78,9±8,3 | 10984±823 |
| | Reproducibility | 79 | 92 | 89 | 93 |
| Nist 614 | Reference values | 3,15 ±0,05 | 45 ±0,4 | 34,39± 0,3 | - |
| | Mean measured values (n=4) | 3,5 ±0,8 | 43,4 ±1,6 | 38,1± 4 | - |
| | Reproducibility | 77 | 96 | 88 | - |
| MACS-1 | Reference values | 114 ±8 | 219±20 | 10 | 110±16 |
| | Mean measured values (n=15) | 110 ±16 | 180±43 | 22 | 326±113 |
| | Reproducibility | 85 | 76 | 85 | 65 |
| MACS-3 | Reference values | 58,7 ±2 | 6760 ±350 | 1756± 136 | - |
| | Mean measured values (n=11) | 47,2 ±12 | 6866 ±420 | 1566± 189 | - |
| | Reproducibility | 74 | 94 | 88 | - |





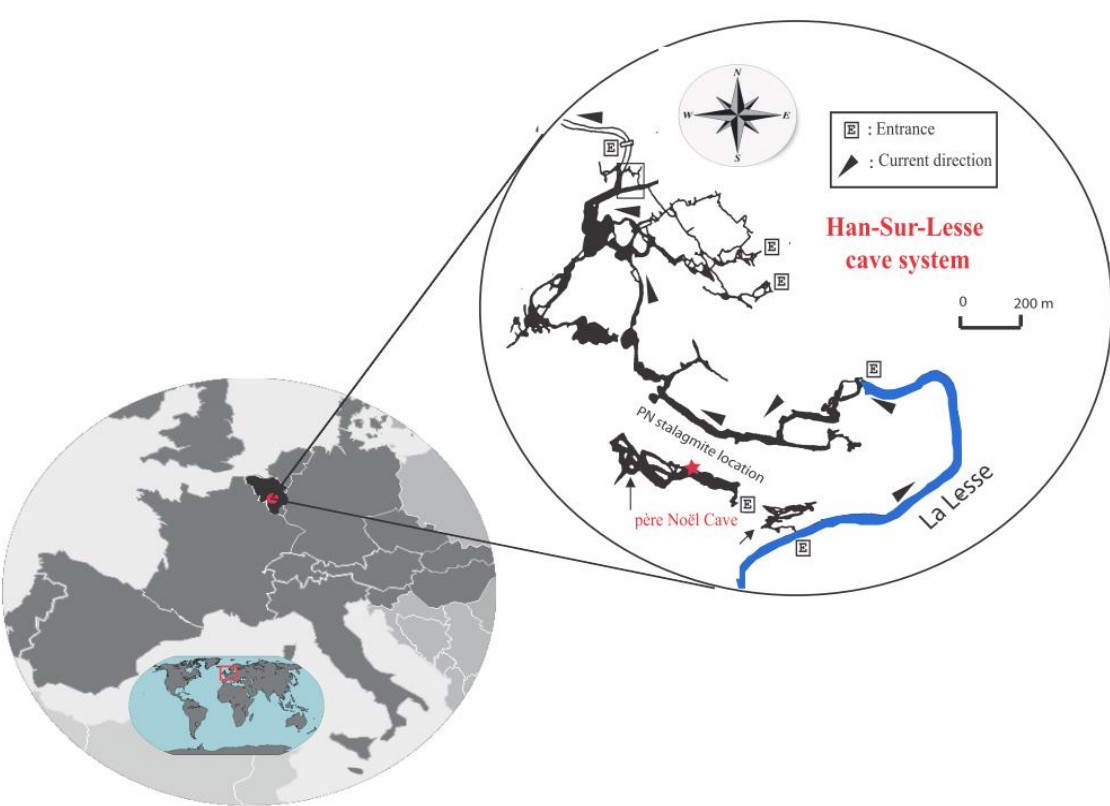

Figure 1: Location of the Père Noël Cave at Han-sur-Lesse, Belgium.



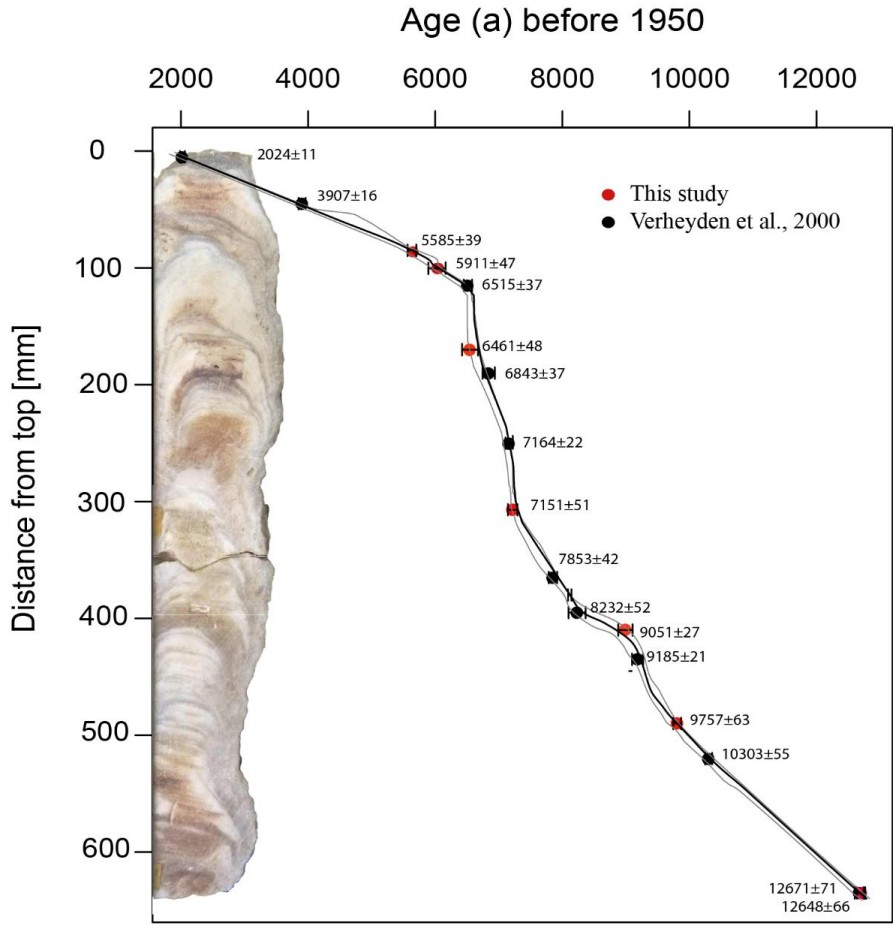

Figure 2: Age versus depth plots and average growth rate of the PN stalagmite. The black line represents the age model. Grey lines indicate the age dating uncertainties. Red and black dots present the U/Th dates.





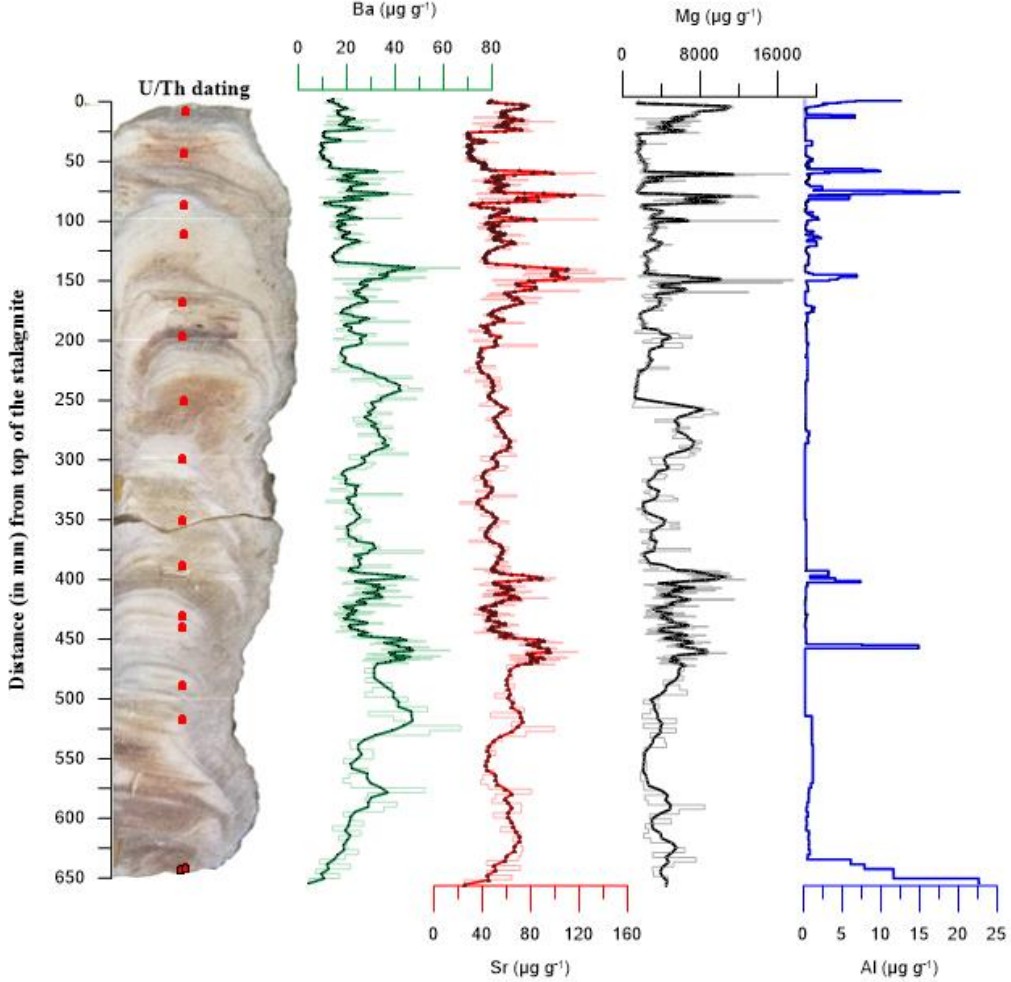

Figure 3: Stalagmite PN laser ablation Sr (red line), Ba (green line), Mg (black line) and Al (blue line) concentrations record for 64 cm-length. The measurement was made with 300-1000 µm intervals (light color). Dark line color presents mean of three measurements.



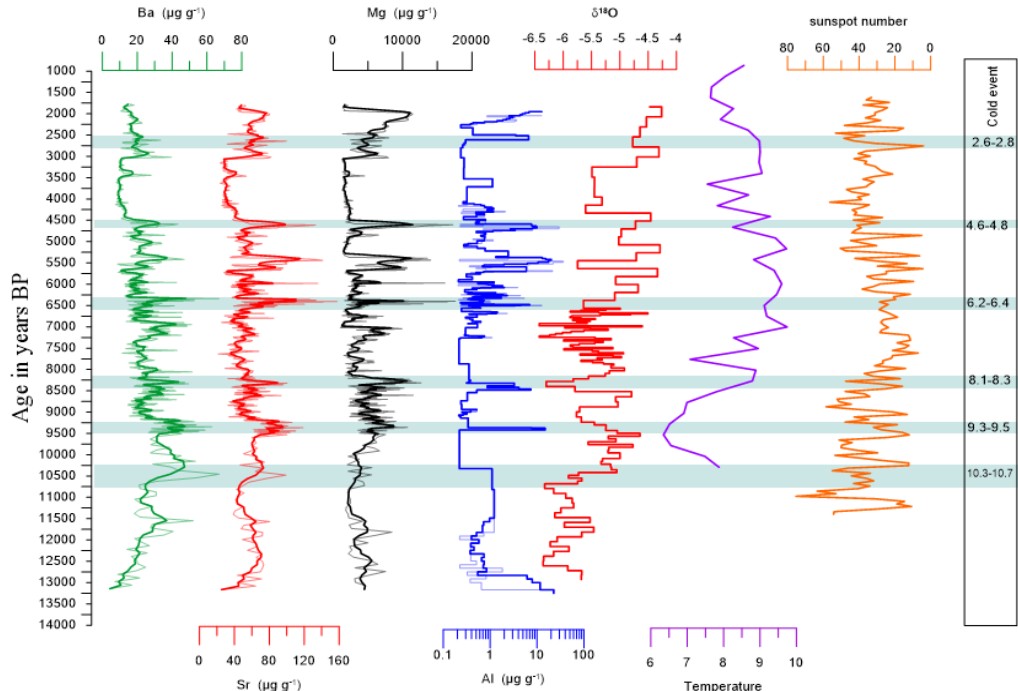

Figure 4: Stalagmite PN laser ablation Sr (red line), Ba (green line), Mg (black line) and Al (blue line) concentrations records for the period 12.7 to 1.8 ka BP. The dark color lines presents mean of three measurements. PN Stalagmite $\delta^{18}$O measurements (red line) were made every 5 mm. Purple line presents the temperature recorded from Mekelermeer core in Netherlands (Bohncke P., 1991). Orange line shows sunspot number related to solar activity (Solanki et al., 2004).





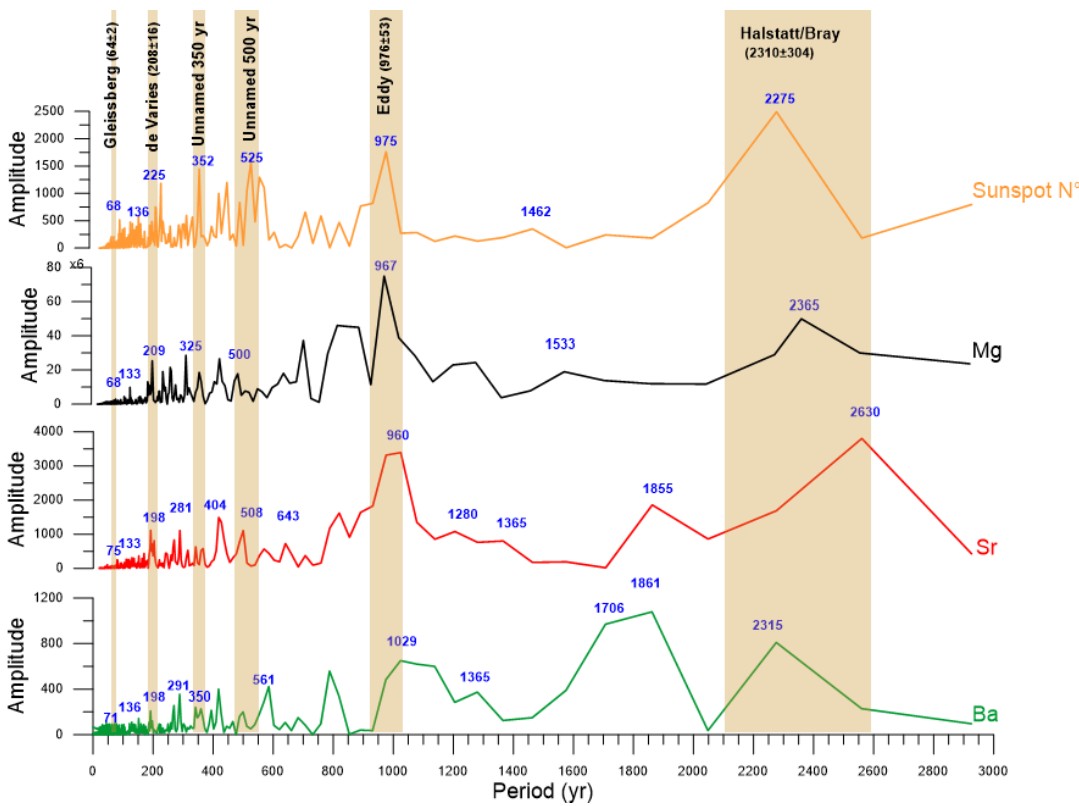

Figure 5: Intensity of cycles in PN trace elements and sunspot activity (Solanki et al., 2004). The power spectra demonstrate that the strongest cycles of PN trace elements are with durations of 68-75, 133-136, 198-209, 291-358, 404-602,912-1029 and 2365-2670 yr.



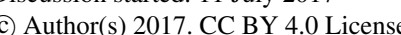

Figure 6: Continuous wavelet transform spectra for sunspot number, temperature, Ba, Sr, Mg, spectral power (variance) is shown by colors ranging from deep blue (weak) to deep red (strong).


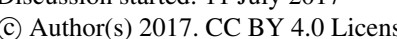


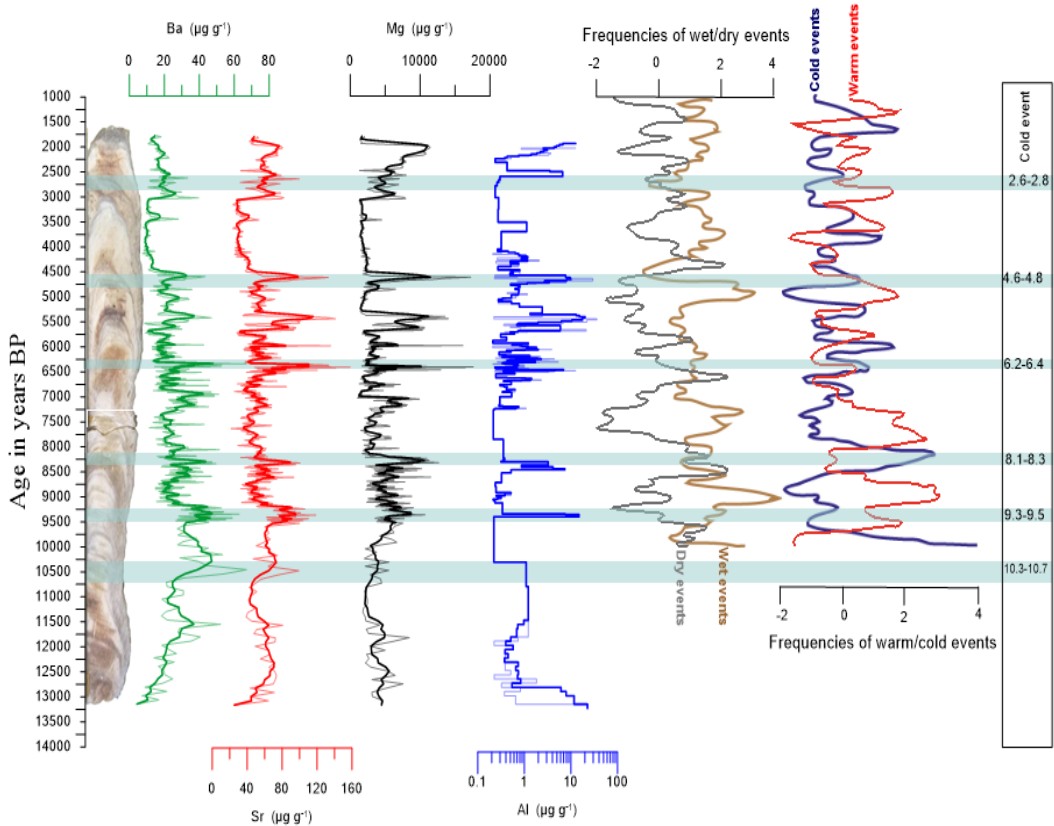

Figure 7: A comparison between PN trace elements contents (Sr = red line; Ba = green line; Mg = black line; and Al = blue line), frequency of warm/cold and wet/dry events in the Northern Hemisphere as defined by Wanner et al. (2014). The blue horizontal bars mark cold events.