# Peer review of "Does Belgian Holocene speleothem records solar forcing and cold events?"

_Climate of the Past, 2017_

## Editor Comment (EC1) · D. Fleitmann (Editor) · 12 Jul 2017

The current title "Does Belgian Holocene speleothem records solar foring and cold events?" contains a grammatical error. Unfortunately, we are not able to correct the title at this stage of the review process. We therefore ask the authors to correct the title of the revised manuscript.

Best wishes, Dominik Fleitmann

---

## Referee Comment (RC1) · Anonymous Referee #1 · 27 Jul 2017

General comment:

This paper addresses an interesting question, namely whether Holocene climate variability was influenced by past changes in solar activity and if these fluctuations are recorded in a speleothem from Belgium.

In general, the paper is well written, the data are interesting and may have the potential to tackle this question. However, in its current form, I cannot recommend the paper for publication in CP. My main concern is that the discussion and interpretation of the presented trace element records is mainly based on correlation and time series analysis. The similarity of the individual trace element signals and their similarity with the d18O record is – in my opinion – not sufficient for a robust interpretation in terms of climate variability. The same is the case for the interpretation in terms of solar forcing. The

simple observation of cycles with a similar period as solar variability may indicate a solar origin, but to in order to prove this relationship, the authors should at least present a hypothetical mechanism, how the trace element records could be influenced by solar variability. In other words, I would like to see a much more detailed discussion of the potential processes controlling the trace element and stable isotope signals of the speleothems.

A few more detailed points are listed below.

Detailed comments:

Section 2 provides a nice summary of the potential processes influencing speleothem trace element signals. This shows that the authors are aware of all these complicated processes. I strongly suggest that they discuss their records in terms of these processes, rather than just relying on correlations and frequency analysis.

Line 114 ff.: The stalagmite was dated in three different labs. In terms of comparability, it would be good to provide a bit more information on the methods used in the individual labs (spike calibration, decay constants, etc.).

Table 1: Please use points instead of commas for decimal separation. In addition, please provide uncertainties for U content and activity ratios. Why are no (230Th/234U) ratios provided for the three samples (NA)? The (230Th/232Th) is relatively low for some samples suggesting significant detrital Th. Have the ages been corrected for this? Is this reflected in the age uncertainties? Please provide more information.

Line 124ff.: Please provide more information on the construction of the age model. Which model was used?

Line 150 and throughout the MS: "Trace element results are closely correlated (r = 0.48-0.77 ..." Please do not provide ranges for r-values, but report the individual correlation coefficients for all records. This information could also be listed in a Table. In addition, how were r-values calculated in case of different temporal resolution of the records?

Interpolation? Please provide more information.

Line 158: Fig. 5 is mentioned prior to Fig. 4 in the text.

Line 163ff.: "The d18O isotopic composition of the calcite in the Père Noël cave is largely controlled by water availability (drip rate) in the cave. Changes in isotopic equilibrium conditions are driven by the changes in cave humidity and linked to changes in precipitation and evapo-transpiration (Verheyden et al., 2008)." Please provide more information on this. Of course, you do not have to repeat all the details if you refer to published work, but the interpretation of the d18O values is crucial for the interpretation of the trace elements. Thus, at least a short discussion of the major drivers of the d18O signal should be given here.

Line 175: "(r(Sr, Ba)=0.77, r(Mg, Sr)=0.72, r(Mg, Ba)=0.48)" This is much better (see above) and should be used throughout the paper.

Line 180ff.: "The positive correlations of Al with Mg, Ba and Sr (r =0.45) suggest that Al content is controlled by the same process than the other trace elements." This is the major problem of the whole paper in its current form. I agree that a correlation may suggest a similar process influencing all trace element records. However, whereas Mg, Sr and Ba are mainly transported as solutes in cave waters, Al is normally transported attached to particles and colloids and thus a proxy for detrital content. Therefore, the correlation of Mg, Sr and Ba with Al – in my opinion – suggests that Mg, Sr and Ba are not only incorporated into the speleothem in dissolved form, but also associated with detrital material. Actually, the positive correlation suggests that the signals are dominated by the content of detrital material. As a consequence, the interpretation of Mg, Sr and Ba in terms of past hydrological variability is complex. For instance, a similarity of these signals has often been interpreted as a result of PCP in the aquifer above the cave. High values would then reflect drier conditions. This could also be proposed for the PN stalagmite. However, high Al content reflecting a high content of detrital material (particles/colloids) would suggest higher flow rates in the aquifer and

thus wetter conditions, which would be contradictory. In summary, as outlined in my general comment, the authors must provide a much more detailed discussion of the processes affecting the trace element and stable isotope records. In the current form, all is based on correlations, which can be – as described above – misleading. In this context, it would also be good to see the d13C record if available. As trace elements are strongly influenced by processes in the soil and karst, the comparison with d13C values my provide important additional information.

Line 187ff.: "First spectral analysis demonstrates that trace element time-series contain significant periodicities ..." Maybe I missed that in the paper, but how was the significance of the periodicities determined?

Line 191 ff.: "Second trace element PN time-series are compared with available temperature record from Mekelermeer core in the Netherlands (Bohncke J., 1991) in order to evidence a regional temperature influence on the PN record." I do not understand why the records are compared with a temperature record if all proxies are interpreted in terms of precipitation. This is a general problem of the paper, in particular in section 5.3. If the records are believed to reflect past precipitation, they should (at least mainly) be discussed in this context. In section 5.3, however, several cold events, such as the 8.2ka event, are discussed as well. Please be more precise here. In addition, it would be better to compare with another record than a relatively old non peer-reviewed record from a PhD thesis.

Line 213ff.: "Since the changes of trace elements in the PN speleothem were formerly demonstrated as due to changes in recharge, i.e. effective precipitation, the study suggests that variations in solar activity may be a significant forcing on either the precipitation or on the soil activity or vegetational activity intensity." If trace elements reflect soil and/or vegetational activity, this could be expressed by a similarity with the d13C signal (see above). I really suggest to show and discuss d13C as well.

Line 221ff.: "The higher positive correlations between trace elements concentrations

(r=0.5-0.9) observed during cold periods ..." See above. Please provide more information here. Which signals were correlated? Which periods were used? Why are cold periods studied if the trace elements reflect precipitation?

Line 234ff.: "The oldest part of the PN stalagmite, from 12.7 to about 12.1 ka BP, is characterized by the lowest contents of trace elements and low d18O, suggesting wet conditions (Fig.4)." Except for d18O (and maybe Ba), I do not see this. Mg and Sr are on a similar level as in many other phases of the Holocene.

Line 260ff.: "The covariance between geochemical proxies support, as explained in Verheyden et al. (2008), their interpretation in terms of dry-wet changes with higher trace elements and higher $\delta$18O values linked to drier conditions." I agree that d18O and the trace element signals show some similarity on the orbital to millennial scale. However, on shorter time scales, and in particular during some of the trace element peaks, d18O values show an opposite behaviour (low d18O/high trace elements, Fig. 4). This suggests that the relationship is not as obvious as proposed by the authors. Please provide a more detailed discussion.

Line 324ff.: "Based on trace element time-series we demonstrate that several events at 10.3, 9.3-9.5, around 8.2, 6.4-6.2, 4.7-4.5, and around 2.7 ka BP alternate with periods of relatively stable and wet/warmer climate." Here and throughout the paper: How do you infer warm/cold, if the proxies reflect precipitation? This needs to be clarified, based on a detailed discussion of the processes influencing the proxy signals ...

---

## Short Comment (SC1) · 19 Aug 2017

The manuscript by Allan et al. under review constitutes a very valuable contribution to an important discussion regarding climate evolution during the Holocene, that could greatly benefit from some improvements. It is exclusively with that aim that I write this commentary.

The paleoclimatology field is undergoing an interesting controversy over the existence of Holocene climatic oscillations or quasi-cycles for which no clear mechanism has been identified. For some authors this type of periodicities are likely to emerge from the noise generated by a chaotic climate system (Turner et al., 2016). To other authors the climatic periodicities are real even if the forcings responsible are unclear at present

(Khider et al., 2014). This manuscript is an important contribution towards cementing the correlation between some of these climate periodicities and solar variability periodicities as recorded by cosmogenic isotopes records. The hydrological nature of the evidence makes it particularly valuable.

The good resolution of the speleothem ($\sim$ 20 years on average), and the high correlation between the different trace elements analyzed make this proxy a very good data source to reconstruct the centennial and millennial climatic changes underlying the differential deposition of the elements.

The article would benefit greatly from a discussion of known hydrological changes during the period analyzed with a special emphasis on how they could affect Père Noël cave hydrology and trace element deposition.

Modeling (Shindell et al., 2001), climate data (Kodera, 2002), and proxy reconstructions (Rimbu et al., 2004) have implicated North Atlantic Oscillation variability as an important element in the climate response to variable solar forcing. Modes of NAO variability are responsible for wind strength and moisture delivery to Europe (Smith et al., 2016). During the periods of lower than average solar forcing discussed in the manuscript, dominant NAO negative conditions caused a southward displacement of moisture delivery to Europe, and are associated with increased humidity in Central and Southern Europe (Smith et al., 2016; Magny, 2004), and decreased humidity in non-coastal Northern Europe (Deininger et al., 2016; Harrison et al., 2002).

The hydrology of the Belgian region, as inferred from the Père Noël cave speleothem record, should be compared to the hydrological groups defined by Harrison et al., 2002, based on lake levels, where it probably belongs to group 5. Even more relevant is the comparison to European speleothem $\delta18O$ records presented by Deininger et al., 2016, as it is the same type of proxy based on a different chemical species. Bunker cave, located in Germany (51°N, 8°W; Fohlmeister et al. 2012) is close enough to merit a comparison that could allow to draw some conclusions on the similitude of

hydrological conditions.

It is also my opinion that the title of the manuscript has been poorly chosen. Speleothems record essentially hydrological changes, and the data presented does not allow any inference towards temperature changes. The paper should concentrate mainly on hydrological changes during the Holocene. There are plenty of papers on Holocene temperatures while papers on hydrology are not so abundant.

I hope my modest contribution is useful both to the authors and the editor of this manuscript.

Bibliography

Deininger, M., McDermott, F., Mudelsee, M., Werner, M., Frank, N., & Mangini, A. (2016). Coherency of late Holocene European speleothem $\delta$18O records linked to North Atlantic Ocean circulation. Climate Dynamics, 1-24.

Fohlmeister, J., Schröder-Ritzrau, A., Scholz, D., Spötl, C., Riechelmann, D. F., Mudelsee, M., ... & Richter, D. K. (2012). Bunker Cave stalagmites: an archive for central European Holocene climate variability. Climate of the Past, 8(5), 1751.

Harrison, S. P., Yu, G., & Vassiljev, J. (2002). Climate changes during the Holocene recorded by lakes from Europe. In Climate development and history of the North Atlantic realm (pp. 191-204). Springer Berlin Heidelberg.

Kodera, K. (2002). Solar cycle modulation of the North Atlantic Oscillation: Implication in the spatial structure of the NAO. Geophysical Research Letters, 29 (8).

Khider, D., Jackson, C. S., & Stott, L. D. (2014). Assessing millennial‐scale variability during the Holocene: A perspective from the western tropical Pacific. Paleoceanography, 29(3), 143-159.

Magny, M. (2004). Holocene climate variability as reflected by mid-European lake-level fluctuations and its probable impact on prehistoric human settlements. Quaternary

international, 113 (1), 65-79.

Rimbu, N., Lohmann, G., Lorenz, S. J., Kim, J. H., & Schneider, R. R. (2004). Holocene climate variability as derived from alkenone sea surface temperature and coupled ocean-atmosphere model experiments. Climate Dynamics, 23(2), 215-227.

Shindell, D. T., Schmidt, G. A., Mann, M. E., Rind, D., & Waple, A. (2001). Solar forcing of regional climate change during the Maunder Minimum. Science, 294 (5549), 2149-2152.

Smith, A. C., Wynn, P. M., Barker, P. A., Leng, M. J., Noble, S. R., & Tych, W. (2016). North Atlantic forcing of moisture delivery to Europe throughout the Holocene. Scientific reports, 6.

Turner, T. E., Swindles, G. T., Charman, D. J., Langdon, P. G., Morris, P. J., Booth, R. K., ... & Nichols, J. E. (2016). Solar cycles or random processes? Evaluating solar variability in Holocene climate records. Scientific reports, 6.

---

## Referee Comment (RC2) · L. Comas-Bru (Referee) · 29 Aug 2017

This manuscript presents a new set of trace element data from a speleothem from Pere Noel's cave. The study is focused in addressing whether solar variability during the Holocene has been recorded in this speleothem.

Even though there is a potential for a very interesting study on trace element variability in speleothems during the Holocene, I cannot recommend this manuscript for publication in its current form.

General comments

My main concern is that the authors do not acknowledge the fact that correlation does not imply causation. The authors base their interpretations solely on correlations and

wavelet analysis, without attempting to invoke the physical mechanisms controlling the observed trace element variability and its potential link with solar variability. Therefore, the scientific methods and assumptions used in this study do not support its main claim: "Our study (...) emphasizes that speleothem trace element profiles may be considered as a new solar activity proxy on decadal to centennial timescales over the Holocene" (L331-333).

Since I do not consider the author's interpretation to be robust, I will not comment on that for now. But I hope to be able to review a revised version of this manuscript at some point!

Specific comments

- The overall presentation of the paper is structured and clear. In this regard, I would only suggest to merge Figure 4 and 7, as most time-series are shown in both figures.

- The English is correct throughout the manuscript (I've listed some typos at the end of this review).

- The significance level of the correlations is not provided for any correlation mentioned throughout the manuscript and therefore the reader does not know whether these are significant or not.

- The authors do not use the age model uncertainties in any of their calculations. They are, however, assessing specific periodicities that may not be significant at all if a Monte Carlo approach is used to take include one of the few uncertainties in the speleothem records that we can quantify!

- No information is provided on how the age model has been constructed. This would be useful to understand the reason behind the asymmetric errors for some sections of the speleothem shown in Fig 2.

- Most of the interpretations are based on wavelet analyses, but information on how they've been constructed is only available in the supplementary data. This should be

part of the main "Methods" section.

- It is not clear how have the authors resampled the trace element data at constant time interval to perform the frequency analyses (has it been with a simple mathematical two-point interpolation?). One needs to be very careful with how the interpolation is done as the reconstructed periodogram (spectral power versus frequency for the resampled series) may differ from the original one. And this would make the reconstructed series unsuitable for wavelet analysis (ie frequencies shown in the wavelet spectrum may be just an artefact, predominantly at high frequencies).

- Regarding the wavelet analyses shown in Fig.6, the authors mention frequencies that I do not manage to see in the plots. For example, in L189, "the temporal stability of those periodicities is confirmed by wavelet results". I cannot see any predominant band (redish colour) in the top three panels (the trace elements) lasting more than about a century over a millennial length record. Therefore, stability of those periodicities is not proven.

- Along the same lines as the comment above, the 1000-yr band mentioned in L269 is only present in Ba but not in Mg or Sr. At the moment the authors mention this band as if it was present in all panels but this difference deserves some discussion.

- Also regarding the wavelet analyses, the authors need to be careful not to interpret frequency signals that are outside of the cone of influence, as these may be under/overestimated. See for example, the discussion on the Early Holocene (section 5.3.1, c. L258 ) where bands outside of the cone are used in the discussion.

- Considering that the three comments below are the same, why do they mention different correlation coefficients? Also, please consider not repeating sentences this similar."Statistically significant positive correlations (r(Sr, Ba)=0.77, r(Mg, Sr)=0.72, r(Mg, Ba)=0.48) are found between Sr, Ba and Mg over the Holocene period suggesting either a common or strongly related controlling process." (L175); "The higher positive correlations between trace elements concentrations (r=0.5-0.9) observed during cold

periods which also suggest common control factors." (L221); "The positive correlations (r=0.47-0.98) were observed between trace elements concentrations during those periods which confirms common control factors." (in L259).

- Assuming that the three main trace elements are controlled by similar mechanisms (which most probably do), the following statements cannot be used to interpret dry conditions: "positive correlation (r=0.45-0.95) was observed between element trace concentrations confirming dryer conditions" (in L290); "The highest positive correlation (r=0.78-0.97) observed between trace element concentrations during this period confirms dry conditions" (in L301).

- The authors say that this is a "high-resolution" record, but no information is provided of the temporal resolution between samples. Moreover, "high-resolution" is a vague term that strongly depends on everyone's expertise and background so, I'd suggest the authors to refrain from using it unless extra information is given.

- I don't understand the following sentence: "I calculated consequently the analysis of 50 micrometers each 300-1000 micrometers interval corresponds to a sample of 1 to 3 years every 0.5 to 50 years" (L126). Please, rephrase.

- How would you explain the peak observed at c. 5750 ka BP (figure 4 and 7), which is higher than some of the other peaks interpreted on the basis of Bond's cycles? (see L221).

- "The larger diameter of the stalagmite compared to its mean diameter suggests that there is enough water to flow down on the flank of the stalagmite and to precipitate calcite" (L236), when? Is this observation enough to interpret a wetter period? Could that precipitation take places many years later?

- In the discussion about the Younger Dryas, the authors present "humid" and "warm" is if they were the same concept (which they're absolutely not!): "All proxies measured in in the PN stalagmite suggest that a humid period occurred between 12.7 and 12.1

ka BP. This is in agreement with the warm climate during the early YD (12.9-12.15 ka BP). . . found in lake sediments from. . ." (from L239 onwards).

- In would be helpful for the reader if the caption of Figure 6 explained the meaning of the y-axis as well as the cone of influence. The authors could consider adding an extra panel of Period vs Power for all series (Mg, Sr, Ba, temperature and Sunspot number). This would help comparing the periodicities of the records.

- L190 and L194: Wavelet analysis is mentioned here as being part of first and third points.

- L180: Figure 3 as well.

Grammatical errors / typos

L89: Treble instead of treble

L40: ". . .and/or. . .., and. . ." in a list of items. Please use and just between the last two.

L126: calculated instead of calculted

L144: ka BP instead of BP ka

L205: there are two commas together after et al.

L301: high instead of highest

Best wishes,

Laia Comas Bru

---

## Short Comment (SC2) · 8 Sep 2017

The high-resolution trace element record and the accurate chronological control of the studied stalagmite definitely invite improved statistical evaluation. I think the study is an interesting contribution and will be a well cited paper after the corrections recommended by the official reviews has been implemented. However, I recognized a few points which were not mentioned by the reviewers and deserve more attention during the revision. I hope this short comment can be useful in the finalization of the revised version.

The minor peaks in the Lomb-Scargle periodograms (LPS) in Fig 5 should be treated cautiously. Actually Lomb-Scargle periodogram analysis estimates the significance

only of the largest peak (e.g. Press et al., 1996) and cannot provide information on the significance of the minor peaks in one round. The significance of the minor peaks can be tested in an iterative way by omitting the highest peak so the signal of the second one can be tested and so on. I mean the signal related to the frequency found to be significant in the LSP could be removed from the record and the analysis can be repeated on the filtered data.

As correctly stated in the supplementary text the unevenly spaced record had to be re-sampled to an evenly spaced record to be applicable for Wavelet analysis. There is only a very short note in the supplementary text saying that this resampling was performed with linear interpolation. This pre-processing step, however, quite crucial and the applied resampling design might affect the variance spectrum especially in the high frequencies (i.e. redden it). A resampling protocol utilizing a spectral control to avoid spectral bias caused by interpolation and retain the original spectral characteristics of the data has been presented recently (Hatvani et al., 2017) which might be useful to the Authors in the revision work. As a related question, no significance level is marked in the Wavelet power spectra of Fig. 6. In lack of significance level it is difficult to evaluate the results. The seminal paper from Torrence & Compo 1998, cited also in the supplementary, provides an excellent guide on the types and estimation techniques of significance levels of Wavelet power spectra.

Some points also need correction related to the solar cycles. The 130-yr periodic signal is called, erroneously, as 'Hale cycle' in the manuscript (e.g. lines 43, 201, and 298 in the original manuscript.) Actually, the 22-yr cycle has been named after George E. Hale as recognition of his great contribution to solar physics. Namely, he found that relating to the reversal of the global magnetic field of the Sun, with the period of 22 years, the polarity of sunspot magnetic fields changes in both solar hemispheres at the start of a 11-year cycle (Hale et al., 1919). Spelling mistakes in the name of the cycles also should be corrected at a few other places in the manuscript e.g: Halstatt (instead of Hallstat) in the abstract. de Vries (instead of de Varies) in section 5.2.

The last sentence in the caption of Fig3 says: Dark line color presents mean of three measurements. Please clarify the sentence. What do you mean? Were there triplicate measurements and the dark line shows their average, or the dark line is the 3-point moving average?

I truly hope to read a revised final version of this interesting contribution in CP soon.

sincerely yours,

Zoltan Kern

Hale, G.E., Ellerman, F., Nicholson, S.B. and Joy, A.H., 1919 The Magnetic Polarity of Sun-Spots. Astrophys. J., 49, 153–178.

Hatvani, I. G., Kern, Z., Leél-Ossy, S., and Demény, A.: Speleothem stable isotope reference records for East-Central Europe – Resampling sedimentary proxy records to get evenly spaced time-series with spectral control, Earth Syst. Sci. Data Discuss., https://doi.org/10.5194/essd-2017-44, 2017.

Press, W.H., Teukolsky, S.A., Vetterling, W.T., Flannery, B.P., 1996. Numerical recipes in C. Cambridge university press, Cambridge. 925p

---

## Author Comment (AC1) · 8 Sep 2017

Sender: M. Allan,
Département de Géologie, Université de Liège
Allée du 6 Août, B-4000 Liège, Belgique
e-mail:mohammedallan@outlook.com

To: Dominik Fleitmann
Editor for Climate of the Past

Object : revised version of paper No: cp-2017-91

September 08, 2017

Dear Professor Dominik Fleitmann

We have the pleasure to prepare for Climate of the Past a revised manuscript entitled " Does Belgian Holocene speleothem record solar forcing? " (No: cp-2017-91). We take the opportunity to submit a revised version of our work, including all reviewer comments. All comments (changing, modifying, spelling) suggested by the reviewers have been taken into account in the manuscript. Here are the answers to the comments of the reviewers on the ms. Note the comments are reported in italic and the reply in blue color. We join after the review the modified ms with changes underlined in yellow color.

Hoping that the revised version fits the requirement for publication in Climate of the Past.

Your sincerely

For the authors,
M. Allan

**D. Fleitmann (Editor)**
*The current title "Does Belgian Holocene speleothem records solar foring and cold events?" contains a grammatical error. Unfortunately, we are not able to correct the title at this stage of the review process. We therefore ask the authors to correct the title of the revised manuscript.*
The title was corrected "Does Belgian Holocene speleothem record solar forcing?"
* * *
**Anonymous Referee #1**

- *General comment:*

*This paper addresses an interesting question, namely whether Holocene climate variability was influenced by past changes in solar activity and if these fluctuations are recorded in a speleothem from Belgium. In general, the paper is well written, the data are interesting and may have the potential to tackle this question. However, in its current form, I cannot recommend the paper for publication in CP. My main concern is that the discussion and interpretation of the presented trace element records is mainly based on correlation and time series analysis. The similarity of the individual trace element signals and their similarity with the d18O record is – in my opinion – not sufficient for a robust interpretation in terms of climate variability. The same is the case for the interpretation in terms of solar forcing. The C1simple observation of cycles with a similar period as solar variability may indicate a solar origin, but to in order to prove this relationship, the authors should at least present a hypothetical mechanism, how the trace element records could be influenced by solar variability. In other words, I would like to see a much more detailed discussion of the potential processes controlling the trace element and stable isotope signals of the speleothems.*
We have taken into account this point. The text has been modified. The relationship between trace elements and solar variability is better explained in the new version of the manuscript. See the following section included in the discussion. Here we present a hypothetical mechanism, how the trace element records could be influenced by solar variability. Climate models and paleoclimate proxies (Shindell et al., 2001; Kodera, 2002; Rimbu et al., 2004) have implicated North Atlantic Oscillation (NAO) variability as an important element in the climate response to variable solar forcing. The NAO is known to influence precipitation $\delta^{18}O$ ($\delta^{18}O_p$) through its control on air temperature and on the trajectory of the westerly winds that carry moisture onto Europe during boreal winters (Smith et al., 2016). During the periods of low solar activity, a NAO negative conditions caused a southward displacement of moisture delivery to Europe, and are associated with increased humidity in Central and Southern Europe (Smith et al., 2016; Magny, 2004). This does not necessarily mean that solar forcing was the main source of all Holocene climate variability since different dynamical processes, such as explosive volcanic eruptions, fluctuations of the ocean thermohaline circulation or internal feedbacks, might also have played an important role (Wanner et al., 2008).

**Detailed comments:**

- *Section 2 provides a nice summary of the potential processes influencing speleothem trace element signals. This shows that the authors are aware of all these complicated processes. I strongly suggest that they discuss their records in terms of these processes, rather than just relying on correlations and frequency analysis.*
The text of the discussion has been expanded. See new paragraph 5.1. $\delta^{18}O$, $\delta^{13}C$ and trace element.

- *Line 114 ff.: The stalagmite was dated in three different labs. In terms of comparability,*
it would be good to provide a bit more information on the methods used in the individual labs (spike calibration, decay constants, etc.).
The methods are detailed in the ms as requested.

- *Table 1: Please use points instead of commas for decimal separation. In addition, please provide*
uncertainties for U content and activity ratios. Have the ages been corrected for this? Is this reflected in the age uncertainties? Please provide more information.
Both the method section and the table have been corrected.

- *Line 124ff.: Please provide more information on the construction of the age model.*
Which model was used?
We use StalAge algorithm, the information and reference have been added in the text.

- *Line 150 and throughout the MS: "Trace element results are closely correlated (r = 0.48-*
0.77 ..." Please do not provide ranges for r-values, but report the individual correlation coefficients for all records. This information could also be listed in a Table. In addition, how were r-values calculated in case of different temporal resolution of the records?

The correlation coefficients are now included in the text. To calculate the coefficient r-values we compare the stable isotopes data measured with a sampling resolution of 5 mm with the low resolution dataset for trace elements (also at 5 mm resolution).

- *Line 158: Fig. 5 is mentioned prior to Fig. 4 in the text.*
We check this: Fig.4 is mentioned in line 149 before Fig.5 (line 158).

- *Line 163ff.: "The d18O isotopic composition of the calcite in the Père Noël cave is largely controlled by water availability (drip rate) in the cave. Changes in isotopic equilibrium conditions are driven by the changes in cave humidity and linked to changes in precipitation and evapo-transpiration (Verheyden et al., 2008)." Please provide more information on this. Of course, you do not have to repeat all the details if you refer to published work, but the interpretation of the d18O values is crucial for the interpretation of the trace elements. Thus, at least a short discussion of the major drivers of the d18O signal should be given here.*
The text has been modified with more explanation. See new paragraph 5.1. $\delta^{18}$O, $\delta^{13}$C and trace element.

- *Line 175: "(r(Sr, Ba)=0.77, r(Mg, Sr)=0.72, r(Mg, Ba)=0.48)" This is much better (see above) and should be used throughout the paper.* Ok

- *Line 180ff.: "The positive correlations of Al with Mg, Ba and Sr (r =0.45) suggest that Al content is controlled by the same process than the other trace elements." This is the major problem of the whole paper in its current form. I agree that a correlation may suggest a similar process influencing all trace element records. However, whereas Mg, Sr and Ba are mainly transported as solutes in cave waters, Al is normally transported attached to particles and colloids and thus a proxy for detrital content. Therefore, the correlation of Mg, Sr and Ba with Al – in my opinion – suggests that Mg, Sr and Ba are not only incorporated into the speleothem in dissolved form, but also associated with detrital material. Actually, the positive correlation suggests that the signals are dominated by the content of detrital material. As a consequence, the interpretation of Mg, Sr and Ba in terms of past hydrological variability is complex. For instance, a similarity of these signals has often been interpreted as a result of PCP in the aquifer above the cave. High values would then reflect drier conditions. This could also be proposed for the PN stalagmite. However, high Al content reflecting a high content of detrital material (particles/colloids) would suggest higher flow rates in the aquifer and thus wetter conditions, which would be contradictory. In summary, as outlined in my general comment, the authors must provide a much more detailed discussion of the processes affecting the trace element and stable isotope records. In the current form, all is based on correlations, which can be – as described above – misleading. In this context, it would also be good to see the d13C record if available. As trace elements are strongly influenced by processes in the soil and karst, the comparison with d13C values may provide important additional information.*
We agree with this comment. The text has been corrected. See new paragraph 5.1. $\delta^{18}$O, $\delta^{13}$C and trace element.

- *Line 187ff.: "First spectral analysis demonstrates that trace element time-series contain significant periodicities ..." Maybe I missed that in the paper, but how was the significance of the periodicities determined?* See supplementary data.

- *Line 191 ff.: "Second trace element PN time-series are compared with available temperature record from Mekelermeer core in the Netherlands (Bohncke J., 1991) in order to evidence a regional temperature influence on the PN record." I do not understand why the records are compared with a temperature record if all proxies are interpreted in terms of precipitation. This is a general problem of the paper, in particular in section 5.3. If the records are believed to reflect past precipitation, they should (at least mainly) be discussed in this context. In section 5.3, however, several cold events, such as the 8.2ka event, are discussed as well. Please be more precise here. In addition, it would be better to compare with another record than a relatively old non peer-reviewed record from a PhD thesis.*
Ok. The text has been modified: in particular we remove the comparison with T° record and the reference has been removed.

- *Line 213ff.: "Since the changes of trace elements in the PN speleothem were formerly demonstrated as due to changes in recharge, i.e. effective precipitation, the study suggests that variations in solar activity may be a significant forcing on either the precipitation or on the soil activity or vegetational activity intensity." If trace elements reflect soil and/or vegetational activity, this could be expressed by a similarity with the d13C signal (see above). I really suggest to show and discuss d13C as well.*
Ok, the 13C data have been added in the new version.

- *Line 221ff.: "The higher positive correlations between trace elements concentrations C4(r=0.5-0.9) observed during cold periods ..." See above. Please provide more information here. Which signals were correlated? Which periods were used? Why are cold periods studied if the trace elements reflect precipitation?*

The text has been modified. The correlation coefficients between the trace elements Sr, Mg and Ba have been calculated and reported in the text for several intervals that are considered as cold events in the literature. In our data set we found higher concentrations in TM during those intervals. We interpret the observed higher concentrations by drier conditions.

- *Line 234ff.: "The oldest part of the PN stalagmite, from 12.7 to about 12.1 ka BP, is characterized by the lowest contents of trace elements and low d18O, suggesting wet conditions (Fig.4)." Except for d18O (and maybe Ba), I do not see this. Mg and Sr are on a similar level as in many other phases of the Holocene.*

We agree. The sentence has been deleted.

- *Line 260ff.: "The covariance between geochemical proxies support, as explained in Verheyden et al. (2008), their interpretation in terms of dry-wet changes with higher trace elements and higher δ18O values linked to drier conditions." I agree that d18O and the trace element signals show some similarity on the orbital to millennial scale. However, on shorter time scales, and in particular during some of the trace element peaks, d18O values show an opposite behavior (low d18O/high trace elements, Fig. 4). This suggests that the relationship is not as obvious as proposed by the authors. Please provide a more detailed discussion.*

We have to be cautious in the comparison between TE and stable isotopes, especially at shorter time scales due to the different sampling resolutions. Stable isotopes have been measured with lower resolution (5 mm) than TE (0.3-1 mm).

- *Line 324ff.: "Based on trace element time-series we demonstrate that several events at 10.3, 9.3-9.5, around 8.2, 6.4-6.2, 4.7-4.5, and around 2.7 ka BP alternate with periods of relatively stable and wet/warmer climate." Here and throughout the paper: How do you infer warm/cold, if the proxies reflect precipitation? This needs to be clarified, based on a detailed discussion of the processes influencing the proxy signals.*

The text has been corrected. We compare the intervals that we interpreted as drier or wetter with events that are reported in the literature and that correspond to warm or cold climate.
* * *
**L. Comas-Bru (Referee 2)**

*This manuscript presents a new set of trace element data from a speleothem from Pere Noel's cave. The study is focused in addressing whether solar variability during the Holocene has been recorded in this speleothem. Even though there is a potential for a very interesting study on trace element variability in speleothems during the Holocene, I cannot recommend this manuscript for publication in its current form.*

**General comments**
*My main concern is that the authors do not acknowledge the fact that correlation does not imply causation. The authors base their interpretations solely on correlations and wavelet analysis, without attempting to invoke the physical mechanisms controlling the observed trace element variability and its potential link with solar variability. Therefore, the scientific methods and assumptions used in this study do not support its main claim: "Our study (. . .) emphasizes that speleothem trace element profiles may be considered as a new solar activity proxy on decadal to centennial timescales over the Holocene" (L331-333).*

The physical mechanisms controlling the observed trace element variability in the Pere Noel cave were previously studied. See the references cited in the text (e.g., Verheyden el al., 2000, 2001, 2008). This is now included in a new paragraph (see lines 180-225).

**Specific comments**
*-The overall presentation of the paper is structured and clear. In this regard, I would only suggest to merge Figure 4 and 7, as most time-series are shown in both figures.*

We prefer to keep the two figures: seven profiles are already presented in Figure 4.

*- The English is correct throughout the manuscript (I've listed some typos at the end of this review).*

Ok, minor errors have been corrected.

*- The significance level of the correlations is not provided for any correlation mentioned throughout the manuscript and therefore the reader does not know whether these are significant or not.*
Ok, more detailed are now given in the method section, the correlation coefficient is significant when r values are close to the value 1: $0.5 < r < 1$: strong correlation, $0 < r < 0.5$: low correlation.

*- The authors do not use the age model uncertainties in any of their calculations. They are, however, assessing specific periodicities that may not be significant at all if a Monte Carlo approach is used to take include one of the few uncertainties in the speleothem records that we can quantify!*
In our study we use the mean age produced by *StalAge*, a program that allows to calculate the range of age at any depth. Figure 2 presents the min, mean and max age for any depth interpolated from StalAge and based on punctual U-Th dates. The age uncertainty, inherent to any paleoclimate record, is influenced by the uncertainty on U-Th dates. Such uncertainty is lower than 71 years (see Table 2) over all the Holocene. Indeed 92% of the time gaps in the original time series are lower than 25 years and the remaining 8% all occur at once at the end of the signal (after 9500 BP). See also supplementary figures 7& 8.

*- No information is provided on how the age model has been constructed. This would be useful to understand the reason behind the asymmetric errors for some sections of the speleothem shown in Fig 2.*
Ok, more detailed are now given in the method section. StalAge (Scholz and Hoffmann, 2011) was used to interpolate the ages between the U/Th-age points.

*- Most of the interpretations are based on wavelet analyses, but information on how they've been constructed is only available in the supplementary data. This should be part of the main "Methods" section.*
We prefer to keep wavelet analyses section the in Supplementary data.

*- It is not clear how have the authors resampled the trace element data at constant time interval to perform the frequency analyses (has it been with a simple mathematical two point interpolation?). One needs to be very careful with how the interpolation is done as the reconstructed periodogram (spectral power versus frequency for the resampled series) may differ from the original one. And this would make the reconstructed series unsuitable for wavelet analysis (ie frequencies shown in the wavelet spectrum may be just an artefact, predominantly at high frequencies).*
Prior to perform the spectral analysis each dataset was resampled as follows : for each age t (in BP) in the evenly spaced sequence 1789, 1799, 1809,..., the corresponding concentration of trace element is obtained through a linear interpolation between two consecutive data points whose ages (in BP) surround t. Since 92% of the time gaps in the original time series are lower than 25 years and the remaining 8% all occur at once at the end of the signal (after 9500 BP), the effects of such a resampling on the results presented are negligible, especially for periods such as the de Vries cycle or larger. See more information in supplementary materials Fig. 6-8.

*- Regarding the wavelet analyses shown in Fig.6, the authors mention frequencies that I do not manage to see in the plots. For example, in L189, "the temporal stability of those periodicities is confirmed by wavelet results". I cannot see any predominant band (redish colour) in the top three panels (the trace elements) lasting more than about a century over a millennial length record. Therefore, stability of those periodicities is not proven.*
Ok, the text has been modified.

*- Along the same lines as the comment above, the 1000-yr band mentioned in L269 is only present in Ba but not in Mg or Sr. At the moment the authors mention this band as if it was present in all panels but this difference deserves some discussion.*
Indeed continuous cyclic period of around 1000 yrs are presented during the interval 10.7-10.3 ka BP but with different spectral powers for the different elements. The 1000-yr band is especially pronounced for Ba and Sr than Mg. The text has been modified.

*- Also regarding the wavelet analyses, the authors need to be careful not to interpret frequency signals that are outside of the cone of influence, as these may be under/overestimated. See for example, the discussion on the Early Holocene (section 5.3.1, c. L258) where bands outside of the cone are used in the discussion.*
No wavelet analyses were discussed in L258. PN speleothem suggests a wet Early Holocene with dryer conditions from 10.7 to 10.3, at 10.0, at 9.7, at 9.2 and from 8.5 to 8.2 ka BP (Verheyden et al., 2014). Trace element contents in the PN stalagmite display significant variability during the Early Holocene, with three maxima observed around 10.5, from 9.5 to 9.2 and around 8.2 ka BP (Fig. 4, 7).

*- Considering that the three comments below are the same, why do they mention different correlation coefficients? Also, please consider not repeating sentences this similar. "Statistically significant positive correlations (r(Sr, Ba)=0.77, r(Mg, Sr)=0.72, r(Mg, Ba)=0.48) are found between Sr, Ba and Mg over the Holocene period suggesting either a common or strongly related controlling process." (L175); "The higher positive correlations between trace elements concentrations (r=0.5-0.9) observed during cold periods which also suggest common control factors." (L221); "The positive correlations (r=0.47-0.98) were observed between trace elements concentrations during those periods which confirms common control factors." (in L259).*

The correlation coefficients between the trace elements Sr, Mg and Ba have been calculated and reported in the text for several intervals :
-all Holocene period (L175),
- for dry/cold periods that centered at 10.5, 9.3, 8.2, 6.3, 5.4, 4.6 and around 2.7 ka BP (L221)
- for Younger  Dryas and Early Holocene(L259).
The repetition has been removed from the text.

*- Assuming that the three main trace elements are controlled by similar mechanisms (which most probably do), the following statements cannot be used to interpret dry conditions: "positive correlation (r=0.45-0.95) was observed between element trace concentrations confirming dryer conditions" (in L290); "The highest positive correlation (r=0.78-0.97) observed between trace element concentrations during this period confirms dry conditions" (in L301).*

We agree. The text has been modified. A paragraph was added to better explain the processes that controls trace elements in PN cave.

*- The authors say that this is a "high-resolution" record, but no information is provided of the temporal resolution between samples. Moreover, "high-resolution" is a vague term that strongly depends on everyone's expertise and background so, I'd suggest the authors to refrain from using it unless extra information is given.*

Ok, it is now added in the method section. The age model reveals a growth rate varied between 0.02 and 0.65 mm.yr$^{-1}$. Consequently, the analysis of a diameter of 50 micrometers each 300-1000 micrometers interval corresponds to a sample of 1 to 3 years every 0.5 to 50 years.

*- How would you explain the peak observed at c. 5750 ka BP (figure 4 and 7), which is higher than some of the other peaks interpreted on the basis of Bond's cycles? (see L221).*

Indeed the peak is observed at 5500 ka and not at 5750 ka BP. See more details in the text for this period. Between 5.5 and 5.4 ka BP, the trace elements display an anticorrelation with sunspot number suggesting relatively dry conditions (Fig.4). During this period, trace elements present frequencies corresponding to different solar cycles, i.e. the Gleissberg cycle (70-100 yr), Hale cycle (130 yr), de Vries cycle (200-210 yr), 500 unnamed cycle and Eddy cycles (1000 yr) (Fig.6).

*- "The larger diameter of the stalagmite compared to its mean diameter suggests that there is enough water to flow down on the flank of the stalagmite and to precipitate calcite" (L236), when? In the lower part at ~12.0 ka. Is this observation enough to interpret a wetter period? Could that precipitation take places many years later?*

The stalagmite diameter is rather large compared to its mean diameter. Dreybrodt (1988, 1999) demonstrated that for constant growth rate of a stalagmite, the stalagmite diameter shows a positive relationship with drip rate and thus water availability (see also chapter 7.2. in Fairchild and Baker (2012) for more explanation.

*In the discussion about the Younger Dryas, the authors present "humid" and "warm" is if they were the same concept (which they're absolutely not!): "All proxies measured in the PN stalagmite suggest that a humid period occurred between 12.7 and 12.1 ka BP. This is in agreement with the warm climate during the early YD (12.9-12.15 ka BP). . . found in lake sediments from. . ." (from L239 onwards).*

We agree. Our chronological and geochemical pattern of the Père Noël stalagmite suggests a humid period. This period is synchronous with warm to mild summer temperatures and the presence of a local vegetation cover during the YD as found in a north-eastern German varved paleolake (Neugebauer et al., 2012).

*- In would be helpful for the reader if the caption of Figure 6 explained the meaning of the y-axis as well as the cone of influence. The authors could consider adding an extra panel of Period vs Power for all series (Mg, Sr, Ba, temperature and Sunspot number). This would help comparing the periodicities of the records.*

Ok. The caption has been modified.

*- L190 and L194: Wavelet analysis is mentioned here as being part of first and third points.*
Ok, it is corrected.

**Grammatical errors / typos**
*L89: Treble instead of treble -* Corrected
*L40: ". . .and/or. . . ., and. . ." in a list of items. Please use and just between the last two.* Modified
*L126: calculated instead of calculated.* Modified
*L144: ka BP instead of BP ka.* Corrected
*L205: there are two commas together after et al.* Corrected
*L301: high instead of highest.* Corrected
* * *
**Interactive comment (J. Vinós)**

*The manuscript by Allan et al. under review constitutes a very valuable contribution to an important discussion regarding climate evolution during the Holocene, that could greatly benefit from some improvements. It is exclusively with that aim that I write this commentary. The paleoclimatology field is undergoing an interesting controversy over the existence of Holocene climatic oscillations or quasi-cycles for which no clear mechanism has been identified. For some authors this type of periodicities are likely to emerge from the noise generated by a chaotic climate system (Turner et al., 2016). To other authors the climatic periodicities are real even if the forcings responsible are unclear at present (Khider et al., 2014). This manuscript is an important contribution towards cementing the correlation between some of these climate periodicities and solar variability periodicities as recorded by cosmogenic isotopes records. The hydrological nature of the evidence makes it particularly valuable. The good resolution of the speleothem (∼ 20 years on average), and the high correlation between the different trace elements analyzed make this proxy a very good data source to reconstruct the centennial and millennial climatic changes underlying the differential deposition of the elements. The article would benefit greatly from a discussion of known hydrological changes during the period analyzed with a special emphasis on how they could affect Père Noël cave hydrology and trace element deposition. Modeling (Shindell et al., 2001), climate data (Kodera, 2002), and proxy reconstructions (Rimbu et al., 2004) have implicated North Atlantic Oscillation variability as an important element in the climate response to variable solar forcing. Modes of NAO variability are responsible for wind strength and moisture delivery to Europe (Smith et al., 2016). During the periods of lower than average solar forcing discussed in the manuscript, dominant NAO negative conditions caused a southward displacement of moisture delivery to Europe, and are associated with increased humidity in Central and Southern Europe (Smith et al., 2016; Magny, 2004), and decreased humidity in non-coastal Northern Europe (Deininger et al., 2016; Harrison et al., 2002). The hydrology of the Belgian region, as inferred from the Père Noël cave speleothem record, should be compared to the hydrological groups defined by Harrison et al., 2002, based on lake levels, where it probably belongs to group 5. Even more relevant is the comparison to European speleothem δ18O records presented by Deininger et al., 2016, as it is the same type of proxy based on a different chemical species. Bunker cave, located in Germany (51°N, 8°W; Fohlmeister et al. 2012) is close enough to merit a comparison that could allow to draw some conclusions on the similitude of hydrological conditions. It is also my opinion that the title of the manuscript has been poorly chosen. Speleothems record essentially hydrological changes, and the data presented does not allow any inference towards temperature changes. The paper should concentrate mainly on hydrological changes during the Holocene. There are plenty of papers on Holocene temperatures while papers on hydrology are not so abundant. I hope my modest contribution is useful both to the authors and the editor of this manuscript.*

Thanks for the positive comments. We therefore modify the title by focusing on solar forcing record in speleothems. The text has been modified and includes now more information about the hydrological conditions. In particular we now present the main results from previous cave monitoring (see section 5.3). The drier intervals that we detect in our PN dataset are compared with European climate event reported in literature. In particular we compare TN profiles with the frequency of warm/cold and wet/dry events as defined by Wanner et al. (2014) in the Northern Hemisphere (see figure 7).

**Does Belgian Holocene speleothem record solar forcing?**

[revised manuscript text omitted]
, following the method described in McDermott et al. (1999). Calcite samples for U/Th dating (0.2–2 g) were spiked with a mixed $^{229}$Th–$^{236}$U spike. U and Th were separated on columns loaded with 2 ml of anion exchange resin (AG1 X8, 200–400 mesh Biorad). The total procedural blanks were negligible (ca. 15 pg of Th and U). Errors

based on counting statistics were < 0.1‰ for all ratios except for $^{230}Th/^{229}Th$, where 2σ varied between 0.4 and 1.0% depending on the U content and age sample. Other dates (7 dates) were done by NEPTUNE Multi-Collector Inductively Coupled Plasma Mass Spectrometry (MC-ICP-MS) at the Laboratoire Géosciences Environnement Toulouse (GET) in France. U and Th were separated from the matrix and purified by ion exchange chromatography using AG1 X8 resin (Riotte and Chabaux, 1999; Violette et al., 2010). Errors based on counting statistics varied between 0.2 and 0.7 % for all ratios. Procedure blanks are <1 pg and ~10 pg for U and Th, respectively, which is negligible compared to the U and Th amounts in samples. A blank is measured between each sample and a standard (HU1) every four samples. All ages were corrected for detrital contamination assuming a $^{232}Th/^{238}U$ mass ratio of 3. One date was obtained in the Earth Science Department of the University of Minnesota on a Thermo-Scientific Neptune-Plus multi-collector inductively coupled plasma mass spectrometer (MC-ICP-MS). The procedures that were followed for uranium and thorium chemical separation and purification described in Edwards et al. (1987) and Cheng et al. (2009a, 2009b). Dating results are summarized in Table 1, with ages given in years before 1950. Errors given in the table (2σ) depend on the U and Th content and age of the sample.

The elementary geochemical composition (Ba, Sr, Mg and Al) of PN stalagmite was determined by Laser Ablation Inductively Coupled Plasma-Mass Spectrometer (LA-ICP-MS) mounted with an ESI New Wave UP-193FX Fast Excimer ArF laser of 193 nm at the Royal Museum for Central Africa (Tervuren, Belgium). Spots were made of 50 µm in diameter and spaced at 300-1000 µm intervals. The age model reveals a growth rate ranging between 0.02 and 0.65 mm.yr$^{-1}$. Consequently, the analysis of a diameter of 50 micrometers each 300-1000 micrometers interval corresponds to a sample of 1 to 3 years every 0.5 to 50 years. Detection limits are calculated from the intensity and standard deviation measurements of the blank. The limits of quantification range between 0.1µg g$^{-1}$ for Sr, Al and Ba, and 2.5µg g$^{-1}$ for Mg. Certified reference materials (NIST 610, NIST 612, NIST 614, MACS-1, and MACS-3) were analyzed with each series of samples, in order to determine the precision and accuracy of analytical procedures. Comparison between reference values and measured values were satisfactory (Table 2) within 65-98 %. For Ba, Sr, and Mg, the reproducibility was higher than 76%. The lowest value was observed for Al (65% for MACS-1 standard). The correlation coefficients (r) allow for the assessment of the degree of linear relationship between two variables. The correlation coefficient is significant when r values are close to the value 1 (0.5< r<1 : strong correlation, 0< r<0.5: low correlation).

To investigate the solar forcing controls on trace elements content in the PN stalagmite, we compared the concentrations of Ba, Sr, and Mg with sunspot number (solar activity-Solanki et al., 2004). Continuous wavelet transform was applied on the trace element time-series data to detect any significant periodicities. Prior to perform the spectral analysis each dataset was resampled as follows : for each age t (in BP) in the evenly spaced sequence 1789, 1799, 1809,..., the corresponding concentration of trace element is obtained through a linear interpolation between the two consecutive data points whose ages (in BP) surround t. Since 92% of the time gaps in the original time series are lower than 25 years and the remaining 8% all occur at once at the end of the signal (after 9.5 ka BP), the effects of such a resampling on the results presented are negligible (especially for periods such as the de Vries cycle or larger). More information in supplementary materials (Fig.6, 7, 8).

**4. Results**

The age model was constructed by using StalAge algorithm (Scholz and Hoffmann, 2011). The 17 ages obtained for stalagmite PN are all in stratigraphic order (Fig. 2). The age-depth model indicates that the PN stalagmite was deposited between 12.7 and 1.8 ka BP. The internal longitudinal section of the stalagmite presents a succession of brown parts and milky white parts along its longitudinal axis (Fig. 2). No clear hiatus is observed in the stalagmite. The longitudinal section also reveals variations in the stalagmite diameter along its longitudinal axis. The growth rate varied between 0.02 and 0.65 mm.yr$^{-1}$ with the highest growth rates (0.09 to 0.65 mm.yr$^{-1}$) observed between 7.8 and 6.3 ka BP (i.e., between 37 and 12 cm). The lowest growth rates evidenced between 6.3 and 1.8 ka BP.

The Ba, Sr, Mg, and Al records in PN are composed of nearly six-hundred independent points. Profiles of Mg, Sr, Ba and Al in PN stalagmite are very similar (Fig. 3). The values (in µg g$^{-1}$) range from 5 to 65 for Ba, 20 to 160 for Sr, 1000 to 17600 for Mg and from 0.2 to 35 for Al. High trace elements contents are observed at different depths: from 540 to 455 mm (10.7-9.2 ka BP), 390 -370 mm (8.2-7.9 ka BP), between 300 and 140 mm (7.2-6.2 ka BP), and from 25 mm (3 ka BP) to the top of PN stalagmite at 2.02 ka BP (Fig.3, 4). In the low trace element content zones, the Al content was below the limit of detection (< 0.1 µg g$^{-1}$). Trace element results are closely correlated ($r_{(Sr, Ba)}$=0.77, $r_{(Mg, Sr)}$=0.72, $r_{(Mg, Ba)}$=0.48, , $r_{(Al, Ba)}$=0.42, $r_{(Al, Mg)}$=0.4g, Fig. 4), implying that similar processes influence their concentration changes, over all timescales. Strontium concentration averages approximately 2-4 times the Ba concentration, throughout the record. Trace element data was resampled at constant time interval to perform spectral analyses (more details in supplementary data). Significant periods of 71, 136, 198, 291, 497-602, 1029, 1365, 1706, 1861 and 2315 are observed in Ba time-series. Strontium also shows significant periodicities at 75, 133, 198, 281, 404-634, 960, 1280, 1365, 1855, and around 2630 yr. In addition, Mg data exhibit peaks around 68, 133, 209, 325, 500, 602-706, 967, 1533 and around 2365 yr. All observed periodicities display some similarities with solar variability, presented by sunspot number, which shows significant cycles at 68, 87, 136, 225, 352, 525, 975, 1462, 2275 yr (Fig.5) .

Stable isotopes data ($\delta^{18}$O and $\delta^{13}$C) from PN stalagmite reported in Verheyden et al. (2008). Stable isotope composition of PN varies between $-4.3$ and $-6.4$‰ for $\delta^{18}$O (mean -5.4‰) and between -2.56 and -9.46‰ (VPDB) for $\delta^{13}$C (mean -7.19 ‰) (Fig. 4). $\delta^{18}$O displays a long term increasing trend from lower values (-6‰) between 12.7 and 10.7 ka BP to -5.3 ‰ between 10.7 and 7.5 ka BP and to higher values (>-5.0‰) between 6.3 and 1.8 ka BP. Lower values are observed at 10.7, between 9.5 and 9, between 8.3 and 6.2, at 5.5 ka BP and between 4.3 and 3.4 ka BP with a general increase until the end of speleothem deposition. The $\delta^{18}$O and $\delta^{13}$C values of PN stalagmite show very similar trends (Fig.4), confirmed by the rather high correlation coefficients of r =0.8. Consequently, we interpret the stable isotope signals recorded by PN stalagmite in terms of past climate variability.

**5. Discussion**

**5.1. $\delta^{18}$O, $\delta^{13}$C and trace element**

In previous studies (Verheyden et al., 2000, 2008; Genty and Deflandre, 1998), the PN stalagmite has received a great deal of scientific attention to 1) understand how climate signals are recorded in proxies speleothems of Père Noël cave (PN); 2) reconstruct climate variations at low resolution of the Holocene . The $\delta^{18}$O and $\delta^{13}$C of PN cave, has been suggested to reflect changes in the importance of evaporation and fast degassing, linked with changes in effective rainfall (Verheyden et al., 2008). To correctly interpret variations in $\delta^{18}$O and $\delta^{13}$C of PN speleothem, continuous monitoring of

several parameters was set up to investigate the controlling factors affecting the $\delta^{18}O$ and $\delta^{13}C$ of PN speleothem. Between 1991 and 1998, different cave parameters were measured such as cave air and seepage water temperature, humidity, $P_{CO2}$, drip water, $\delta^{18}O$ and $\delta^{13}C$ of seepage water, conductivity, trace element concentrations of the water and calcite. Several statements are derived from this monitoring:

(1) The mean annual precipitation of the nearby meteorological station (Lessive, located at 3 km NE of the cave), measured over the period 1980-1998, is 826 mm. The mean temperature inside the cave of 9 °C (varies between 8.5° and 9.2 °C) corresponds to the mean annual temperature of the air (9.2°C) at the nearby Lessive station.

(2) The relative humidity changes from 95 to 98%. There is a good correlation between the water excess (or rainfall), and the annual flowrate of the stalactite ($R^2$= 0.98). It demonstrates that the quantity of water feeding a stalagmite is controlled by the annual rainfall.

(3) the mean $\delta^{18}O$ of seepage water of -7.5‰ (-8.0 <$\delta^{18}O$ < -6.8 ‰) which is similar to annual mean rain water (-7.3‰). More negative $\delta^{18}O$ values of calcite occur during periods of higher cave effective precipitation when calcite deposition occurs closer to isotopic equilibrium.

(4) The stable isotopes signals ($\delta^{18}O$ and $\delta^{13}C$) were interpreted as due to variation in cave humidity and drip rate producing changes in the kinetics of the calcite deposition. Hendy tests indicated PN stalagmite was deposited out of equilibrium. The enrichment in both $\delta^{18}O$ and $\delta^{13}C$ in calcite and their positive and significant correlation are signs for a non-equilibrium precipitation.

(5) $\delta^{18}O$ and $\delta^{13}C$ of the stalagmite may give some information on dry/wet variations in the past: more negative $\delta^{18}O$ and $\delta^{13}C$ values occur during periods of higher cave water recharge, when calcite deposition occurs closer to isotopes equilibrium).

In this study, statistically significant positive correlations ($r_{(Sr, Ba)}$=0.77, $r_{(Mg, Sr)}$=0.72, $r_{(Mg, Ba)}$=0.48) are found between Sr, Ba and Mg over the Holocene period suggesting either a common or strongly related controlling process. Changes in element traces content in PN stalagmite are interpreted as due to changes in water residence time linked to changes in water availability. Higher Mg and Sr contents were previously observed during dry periods, characterized by longer water residence times (Verheyden et al., 2008) and associated to lower water availability. Since water residence time is related with water availability, trace elements in PN speleothem probably register changes in effective precipitation, i.e., precipitation minus evapotranspiration. The $\delta^{18}O$ and $\delta^{13}C$ and trace elements (Sr, Ba, and Mg) profiles show very similar trends, which is confirmed by the rather high correlation coefficients (r= 0.45-0.70), suggesting a common control. Previous studies (Verheyden et al., 2008) on PN stalagmite showed a similarity between $\delta^{18}O$ and $\delta^{13}C$ and Mg and Sr time-series, which may explain by two processes. First, Mg and Sr contents depend on their concentrations in the seepage waters and on the residence time of the water in the vadose zone, and consequently, through the recharge amount, on the rainfall amount. The dependence on the residence time might be due to the degree of "prior calcite precipitation" (PCP) that will increase both Mg and Sr in the seepage water and consequently in the calcite. $\delta^{18}O$ and $\delta^{13}C$ contents are controlled by the isotopic equilibrium state during precipitation of the calcite, which depends on the drip rate, and consequently, through recharge, on the rainfall amount too. Second, both chemical and isotopic compositions of the stalagmite depend on kinetic processes during calcite precipitation, which are linked with drip rate and water recharge. Consequently, chemical and isotopic ratios appear to be two proxies for the rainfall amount. In the PN speleothem, the

crossed link of stable isotopes ($\delta^{18}$O and $\delta^{13}$C) as of trace element contents (Sr, Ba, and Mg) to water recharge may explain the covariation of the geochemical parameters over much of the Holocene period. In addition, this relationship may also be the consequence of kinetic effects during calcite precipitation as suggested by the observed similar variations of three parameters ($\delta^{18}$O and $\delta^{13}$C and trace elements) along a single layer of the Holocene stalagmite.

Aluminum can be used to determine the variations in the detrital (non-carbonate) content in speleothems. These particles may be transported attached to particles and colloids during periods of intense precipitation, which results in high drip rates (White, 1997; Schimpf et al., 2011). In PN stalagmite, in the low trace element content zones, the Al content was below the limit of detection ($< 0.1$ µg g$^{-1}$) (from 10.2 to 9.3 ka, between 8.1 and 7.2 ka and from 4.1 to 2.8 ka BP; Fig.3, 4). However, Al concentration was higher than the detection limit during the cold events and its maximum values coincide with maxima in the other investigated trace elements (Fig.4). High Al content reflecting a high content of detrital material (particles/colloids) would suggest higher flow rates in the aquifer and thus wetter conditions. In contrast high values of Sr, Ba and Mg reflect drier conditions. This contradiction can be explain as follows as: during the drier periods trace elements (Sr, Ba, and Mg) continue to accumulate in the vadose zone, where the residence time is long, then a high flow rate flashed trace elements and leaded them to precipitate in the cave. During wetter periods, the high drip rates may be transport Al "attached to particles and colloids" from the soil and the trace elements from vadose zone to the cave environment.

**5.2. Spectral and wavelet analyses**

First spectral analysis demonstrates that trace element time-series contain dominant periodicities on decadal to millennial times scales (Fig.5). The most frequently occurring periodicities were those at 68-75, 133-136, 198-209, 291-358, 404-602,912-1029 and 2365-2670 yr (Fig.5). Those periodicities are confirmed by wavelet results (Fig. 6). Second we test PN time-series for solar forcing by applying wavelet approach. Solar forcing was represented by the number of sunspots (Solanki et al., 2004). Wavelet analyses of Ba, Sr and Mg time-series with reconstructed sunspot number time series reveal significant common periodicities within (63–80, 133-140, 198–220, 514-561, 912-1029 and 2245-2670 yr-Fig.6, supplementary data). Those periodicities coincide with known solar cycles (e.g., Steinhilber et al., 2012; McCracken et al., 2013) and are in agreement with decadal-centennial scale variability in Holocene climatic records from widely dispersed geographic regions. For example, the 63–80 yr interval is similar to the Gleissberg cycle (70-100 yr) whereas the 133-140 yr frequency interval corresponds to the Hale cycle (130 yr). The 198–220 yr cycle of PN record is close to the de Vries (200-210 yr) solar cycle (Steinhilber et al., 2012). De Varies's cycle was already observed in numerous palaeoclimate records all over the world (Wanger et al., 2001; Duan et al., 2014; Czymzik et al., 2016). There are also strong variability around 281-325 yr and 535 yr (514-561) periods that correspond to solar variability that reported as "350 and 500 unnamed cycles" (Steinhilber et al., 2012). These cycles were also detected in numerous other palaeoclimatic records worldwide (e.g., Soon et al., 2014; Chapman and Shackleton, 2000). The 960-1029 yr cycle of the speleothem could match the Eddy cycle (1000 yr) as recognized in both ice cores (Stuiver et al., 1995) and marine sediments (e.g., Chapman and Shackleton, 2000). The trace elements PN records show small peaks between 1280 and 1533 yr (Fig. 5). This periodicity is identical to the Bond cycle (1470 ± 500 yrs) detected from ice rafted debris in North Atlantic sediment cores (Bond et al., 2001). Finally the 2245-2670 yr cycle, could be related to the Bray or Halstatt cycle (2200-2400 yr) that is recognized in other palaeoclimatic records (e.g., kern et al., 2012). The significant common periodicities between trace elements PN records and Solar records suggest that solar variability

influenced PN trace elements content on decadal to millennial time scales. Since the changes of trace elements in the PN speleothem were formerly demonstrated as due to changes in recharge, i.e. effective precipitation, the study suggests that variations in solar activity may be a significant forcing on either the precipitation or on the soil activity. Climate models and paleoclimate proxies (Shindell et al., 2001; Kodera, 2002; Rimbu et al., 2004) have implicated North Atlantic Oscillation (NAO) variability as an important element in the climate response to variable solar forcing. The NAO is known to influence precipitation $\delta^{18}O$ ($\delta^{18}O_p$) through its control on air temperature and on the trajectory of the westerly winds that carry moisture onto Europe during boreal winters (Smith et al., 2016). During the periods of low solar activity, NAO negative conditions caused a southward displacement of moisture delivery to Europe, and are associated with increased humidity in Central and Southern Europe (Smith et al., 2016; Magny, 2004). However, this does not necessarily imply that solar forcing was the main source of all Holocene climate variability since different dynamical processes, such as explosive volcanic eruptions, fluctuations of the ocean thermohaline circulation or internal feedbacks, might also have played an important role (Wanner et al., 2008).

[revised manuscript text omitted]

---

## Author Comment (AC2) · 8 Sep 2017

The comment was uploaded in the form of a supplement:
https://www.clim-past-discuss.net/cp-2017-91/cp-2017-91-AC2-supplement.pdf

———————————————

---

## Author Comment (AC5) · 12 Sep 2017

Dear Dr. Zoltan,

Thank you for your interest and comments. Here are reported our answers (in blue color).

*The minor peaks in the Lomb-Scargle periodograms (LPS) in Fig 5 should be treated cautiously. Actually Lomb-Scargle periodogram analysis estimates the significance C1 CPD Interactive comment Printer-friendly version Discussion paper only of the largest peak (e.g. Press et al., 1996) and cannot provide information on the significance of the minor peaks in one round. The significance of the minor peaks can be tested in an iterative way by omitting the highest peak so the signal of the second one can be tested and so on. I mean the signal related to the frequency found to be significant in the LSP could be removed from the record and the analysis can be repeated on the filtered data*

We agree with you that the minor peaks in the Lomb-Scargle periodograms should be treated cautiously.

*As correctly stated in the supplementary text the unevenly spaced record had to be re-sampled to an evenly spaced record to be applicable for Wavelet analysis. There is only a very short note in the supplementary text saying that this resampling was performed with linear interpolation. This pre-processing step, however, quite crucial and the applied resampling design might affect the variance spectrum especially in the high frequencies (i.e. redden it). A resampling protocol utilizing a spectral control to avoid spectral bias caused by interpolation and retain the original spectral characteristics of the data has been presented recently (Hatvani et al., 2017) which might be useful to the Authors in the revision work.*

It was already noticed by one reviewer and it is now corrected in the text. In the revised version, we explain the resampling protocol utilized in our ms. Prior to perform the spectral analysis each dataset was resampled as follows : for each age t (in BP) in the evenly spaced sequence 1789, 1799, 1809,..., the corresponding concentration of trace element is obtained through a linear interpolation between two consecutive data points whose ages (in BP) surround t. Since 92% of the time gaps in the original time series are lower than 25 years and the remaining 8% all occur at once at the end of the signal (after 9500 BP), the effects of such a resampling on the results presented are negligible, especially for periods such as the de Vries cycle or larger. See more information in supplementary materials Fig. 6-8.

*As a related question, no significance level is marked in the Wavelet power spectra of Fig. 6. In lack of significance level it is difficult to evaluate the results. The seminal paper from Torrence & Compo 1998, cited also in the supplementary, provides an excellent guide on the types and estimation techniques of significance levels of Wavelet power spectra.*

In figure 6, continuous wavelet transform spectra for sunspot number, Ba, Sr, Mg. Spectral power (variance) is shown by colors ranging from deep blue (weak) to deep red (strong). Ba, Sr, and Mg concentrations and *sunspot number are presented along y-axis. The white contour lines represent 95% confidence level. The caption text has been modified.*

*Some points also need correction related to the solar cycles. The 130-yr periodic signal is called, erroneously, as 'Hale cycle' in the manuscript (e.g. lines 43, 201, and 298 in the original manuscript.) Actually, the 22-yr cycle has been named after George E. Hale as recognition of his great contribution to solar physics. Namely, he found that relating to the reversal of the global magnetic field of the Sun, with the period of 22 years, the polarity of sunspot magnetic fields changes in both solar hemispheres at the start of a 11-year cycle (Hale et al., 1919).*

The 130 yr frequency interval corresponds to the subharmonics of the Hale cycle. See the following references: *Attolini, M. R., Cecchini, S., Galli, M. & Nanni, T.: On the persistence of the 22 y solar cycle. Sol. Phys. 125, 389–398, 1990; Tuner, T.E., Swindles, G.T., Charman, D.J., Langdon, P.G., Morris, P.J., Booth, R.K., Parry, L.E., Nicolas, J.E.: Solar cycles or random processes? Evaluating solar variability in Holocene climate records. Sci. Rep. 6, 23961; doi: 10.1038/srep23961 (2016).*

*Spelling mistakes in the name of the cycles also should be corrected at a few other places in the manuscript e.g: Halstatt (instead of Hallstat) in the abstract. de Vries (instead of de Varies) in section 5.2.* Ok, it is now corrected.

*The last sentence in the caption of Fig.3 says: Dark line color presents mean of three measurements. Please clarify the sentence. What do you mean? Were there triplicate measurements and the dark line shows their average, or the dark line is the 3-point moving average?*

Indeed the Dark line color in Fig. 3 corresponds to the 3-point moving average.